# Thompson Sampling For Bandits With Cool-Down Periods

**Jingxuan Zhu**                                                      *zhujx5@chinatelecom.cn*
*E-Surfing Digital Life Technology Co., Ltd.*
*China Telecom*

**Bin Liu**[*]                                                        *bins@ieee.org*
*E-Surfing Digital Life Technology Co., Ltd.*
*China Telecom*

**Reviewed on OpenReview:** *https://openreview.net/forum?id=1fvOZS2mXm*

## Abstract

This paper investigates a variation of dynamic bandits, characterized by arms that follow a periodic availability pattern. Upon a "successful" selection, each arm transitions to an inactive state and requires a possibly unknown cool-down period before becoming active again. We devise Thompson Sampling algorithms specifically designed for this problem, guaranteeing logarithmic regrets. Notably, this work is the first to address scenarios in which the agent lacks knowledge of each arm's active state. Furthermore, the theoretical findings extend to the sleeping bandit framework, offering a notably superior regret bound compared to existing literature.

## 1 Introduction

Within the realm of multi-armed bandits (MAB), agents navigate the delicate balance between exploration and exploitation to maximize cumulative rewards. The MAB framework has been applied to a diverse array of practical scenarios, including but not limited to dynamic pricing (Misra et al., 2019), online learning of dialogue systems (Liu et al., 2018) and anomaly detection (Ding et al., 2019). Our research is focused on the stochastic multi-armed bandit problem, a fundamental variant within this domain, where rewards for each action are drawn from a stochastic distribution that is unknown to the agent. The agent's objective is to discern and select the arm that promises the highest average reward over time.

The literature has introduced a spectrum of algorithms to tackle this problem within the standard framework, prominently featuring the Upper Confidence Bound (UCB) algorithm suite (Auer et al., 2002; Garivier & Cappé, 2011; Kaufmann et al., 2012), alongside "greedy" approaches (Langford & Zhang, 2007; Vermorel & Mohri, 2005) and Thompson sampling method (Thompson, 1933). Additionally, foundational work by (Lai et al., 1985) establishes a theoretical lower bound for the performance in these contexts. In recent years, the research focus has shifted towards multi-agent variations of these problems, characterized by distinctions in agent behavior (cooperative or competitive) (Zhang et al., 2021) and how information exchange is executed (Krishnamurthy & Wahlberg, 2009). Notable advancements include distributed cooperative bandit settings (Bistritz & Leshem, 2018; Landgrena et al., 2021; Zhu et al., 2021; Martínez-Rubio et al., 2019; Zhu & Liu, 2023), where multiple agents collaboratively achieve objectives through distributed communication networks; the "collision" setting (Nayyar et al., 2016; Liu & Zhao, 2010; Bistritz & Leshem, 2018; Lai et al., 2008), where agents "collide" when they select a same arm at the same time; and federated information structures (Shi & Shen, 2021; Shi et al., 2021; Zhu et al., 2021; Dubey & Pentland, 2020; Li et al., 2020; Réda et al., 2022), which explore personalized and federated strategies for leveraging shared knowledge.

Despite the rich body of literature, there is a prevailing assumption that the set of arms and their reward distributions remain unchanged over time, a simplification that often diverges from the dynamic nature of decision-making in real-world scenarios. Real-life applications frequently feature changing options and reward distributions, leading to the emergence of a new topic: "dynamic bandit" or "non-stationary bandit"

---

* Corresponding author

problems. The literature encompasses a variety of these scenarios, including the restless bandit framework where arm states evolve based on an irreducible Markov chain (Ortner et al., 2012), scenarios with a finite number of distribution switches (Kocsis & Szepesvári, 2006; Garivier & Moulines, 2011; Auer et al., 2019), and the "sleeping bandit" problem where only a subset of arms is active at any given time (Kleinberg et al., 2010; Chatterjee et al., 2017).

To address the dynamic bandit problem while maintaining logarithmic regret, our work builds upon the "sleeping bandit" framework, which accounts for the dynamic availability of options. To generalize this setting, we introduce a novel extension by incorporating a structured and predictable pattern in the distributional changes. Specifically, we propose a model where each option, or arm, is assigned a unique "cool-down" period, which may not be explicitly revealed to the agent. Upon a "successful" selection of an arm, it becomes temporarily inactive and only yield 0-rewards. This model mirrors real-world scenarios where options are not continuously available, offering a more nuanced approach to the exploration-exploitation dilemma. This setup finds relevance in situations where choices exhibit periodic accessibility, such as:

**Cognitive Radio Networks** In this scenario, a cognitive radio (Wang & Liu, 2010; Masonta et al., 2012) aims to assign the most suitable active channel (arm) for a sequence of communication demands. Each channel possesses a distinct success rate for successful communication. Upon a communication passing through a channel, it becomes occupied for a certain period (cool-down period), during which selecting it again would lead to a "collision" (Liu & Zhao, 2010), resulting in the failure of subsequent communication attempts. To devise an optimal allocation strategy, the agent must effectively acquire knowledge about both the success rate and the cool-down duration of each channel.

**Coupon Website** In this context, we consider a platform that collects promotional vouchers for shoppers. The availability of these vouchers to any given customer is subject to both cool-down constraints and targeting range limitations. These vouchers may only be redeemed once per day or week due to cool-down limitations, and they are distributed to invited users in specific ratios. The goal of the website is to maximize its aggregate positive feedback, measured by the successful redemption of coupons (positive feedback), whereas attempts to use a voucher during its cool-down period or by users not targeted by the promotion result in no reward (negative feedback).

**Gaming** In gaming scenarios, whether against computers or other players, certain events or abilities are designed with cool-down times to ensure fair competition. Additionally, different abilities yield varying outcomes. A player, seeking to secure victories and maximize the reward, must make strategic decisions about which abilities to use and their timing, based on the cool-down durations and the specific benefits of each ability.

## 1.1 Contributions

This paper delves into a special dynamic bandit problem where each arm undergoes a cool-down period following a "successful" selection. The main contributions can be summarized as follows.

1. We present distinct Thompson-sampling based algorithms tailored for scenarios where the cool-down durations are either known or unknown to the agent. In both instances, we establish an upper bound of order $O(\log T)$ for the regret.

2. To the best of our knowledge, this is the first paper that provides a solution to the bandit problem where the active state of each arm is unknown to the agent (the unknown cool-down duration scenario). Besides, when the agent is aware of the cool-down durations, our regret bounds boil down to the results in traditional bandit settings disregarding cool-down periods.

3. Additionally, we extend our algorithm to the sleeping bandit framework, achieving significant improvements compared to existing literature (Kleinberg et al., 2010; Chatterjee et al., 2017).

## 1.2 Problem Formulation

Let us consider a situation where an agent is presented with $M$ distinct arms or options. Each arm can be in one of two states at any given time: active or inactive. Even when inactive, an arm can still be pulled but would yield a 0-reward. The set of arms that are active at time $t$ is represented by $\mathcal{A}(t)$. For every discrete instance $t$, the agent selects an arm $a(t)$, guided by a predetermined policy, thereby receives

a reward, denoted as $X^{(a(t))}(t)$. For any given arm $k$, where $k = 1, \ldots, M$, and at each time $t$, should arm $k$ be active, its reward $X^{(k)}(t)$ is governed by a Bernoulli distribution characterized by an unknown success rate $p^{(k)}$ within the interval $(0, 1)$. This distribution can be represented as follows:

$$X^{(k)}(t) = \begin{cases} 1 & \text{with probability } p^{(k)} \\ 0 & \text{with probability } 1 - p^{(k)} \end{cases}, \quad \text{given } k \in \mathcal{A}(t).$$

Without loss of generality, it is assumed for simplicity that the success rates are sorted in descending order: $p^{(1)} > p^{(2)} > \cdots > p^{(M)}$. When $k$ is inactive at time $t$, $X^{(k)}(t) = 0$.

An arm, upon being chosen and yielding 1-reward (succeeding in the Bernoulli trial) at time $t$, transitions into an inactive state, necessitating a cool-down period of length $l^{(k)}$ before it is reactivated. This inactivity period is mathematically formulated as:

$$k \notin \mathcal{A}(\tau), \quad \text{for } \tau = t + 1, \ldots, t + l^{(k)},$$
$$k \in \mathcal{A}(\tau), \quad \text{for } \tau = t + l^{(k)} + 1.$$

Here, $l^{(k)}$ denotes the cool-down duration for arm $k$, which is constrained by a uniform upper bound $D$, such that $l^{(k)} \leq D$ for any arm $k = 1, \ldots, M$. We explore both scenarios in this paper where the cool-down duration $l^{(k)}$ is either known or unknown to the agent, with the agent being cognizant of the upper bound $D$ in situations where $l^{(k)}$ remains undisclosed.

The agent's primary objective is to minimize its regret, which is defined as the expected cumulative "loss" in reward resulting from not choosing the optimal arm at every instance $t$, and is mathematically described as:

$$R(T) = \sum_{t=1}^{T} (p^{(a^*(t))} - p^{(a(t))}),$$

where $a^*(t) = \arg\max_{i \in \mathcal{A}(t)} p^{(i)}$ identifies the optimal arm at time $t$, which is with the highest success rate among all active arms.

### 1.3 Technical Challenges

In this section, we mainly present the technical challenges when the cool-down duration of each arm is unknown to the agent. Addressing the problem requires the agent to thoroughly explore both the success rate and the cool-down duration for each arm. This exploration faces three significant technical hurdles:

**Limited Exploration of Distributions:** Insufficient exploration of the success rates of each arm may lead to inaccurate estimations by the agent. Consequently, the agent may repeatedly select a sub-optimal arm, resulting in considerable regret.

**Inadequate Exploration of Cool-Down Durations:** Insufficient exploration on the cool-down duration can result in two types of errors: (1) the agent may erroneously choose an inactive arm that it believes to be active, or (2) it might overlook the optimal active arm due to a mistaken belief that it is inactive. Both errors lead to sub-optimal selections. Particularly in the first scenario, incorporating the 0-reward obtained due to selecting an inactive arm into its success rate estimation could bias the estimate, potentially causing the regret to escalate linearly. We demonstrate in Section 6.2 the repercussions of not implementing a cool-down exploration strategy for this bandit problem.

**Over-Exploration of Cool-Down Durations:** In the process of validating the cool-down duration for an arm, the agent may deliberately select an arm presumed to be inactive to gauge the precision of its cool-down estimate. Consequently, this procedure mandates selecting an inactive arm at least once during each verification attempt, inherently resulting in loss in reward as a direct consequence of cool-down exploration. That is why excessive exploration of cool-down duration can exacerbate regret. Further elaboration on the ramifications of over-exploration of the cool-down durations is provided in Section 6.4.

In conclusion, the agent must navigate both distribution (success rate) exploration and cool-down exploration simultaneously. This necessitates a delicate equilibrium between the two explorations. Moreover, to mitigate

the risk of sub-optimal selections stemming from excessive cool-down exploration, it is essential to devise a strategy enabling the agent to strategically pause or resume cool-down exploration based on its confidence level in the cool-down estimate.

## 2 Algorithms

To simplify the presentation, we generally exclude the factor $t$ from most notations used in the pseudocode in this section and Appendix B.

---

**Algorithm 1:** Known Cool-Down Durations

---

**Input:** total time $T$, upper bound $D$

**1 Initialization** For $k = 1, \ldots, M$, set $\alpha^{(k)} = \beta^{(k)} = 1$. Set $t = 0$.

**2 while** $t < T$ **do**

**3**      $t = t + 1$

**4**      **for** $k \in \mathcal{A}$ **do**

**5**          The agent samples $\theta^{(k)}$ from $\text{Beta}(\alpha^{(k)}, \beta^{(k)})$

**6**      **end**

**7**      The agent selects arm $a(t) = \arg\max_{k \in \mathcal{A}} \theta^{(k)}$             `// Thompson Sampling decision-making`

**8**      **if** $X_{a(t)} = 1$ **then**

**9**          $\alpha^{(a(t))} = \alpha^{(a(t))} + 1$

**10**     **else**

**11**        $\beta^{(a(t))} = \beta^{(a(t))} + 1$

**12**     **end**

**13 end**

---

### 2.1 Known Cool-Down Duration

When the cool-down duration of each arm is known to the agent, the problem is equivalent to finding the optimal arm within a given subset $\mathcal{A}(t)$ at each time $t$. To achieve this, we made slight adjustments to the Thompson Sampling algorithm, sampling only from the active arm set rather than the entire arm set at each time step, as outlined in Algorithm 1.

### 2.2 Unknown Cool-Down Duration

When the cool-down duration of each arm is unknown to the agent, as discussed in Section 1.3, it becomes crucial to strike a balance between cool-down exploration and distribution exploration. The agent must first determine which arm to select and decide the type of exploration to undertake before executing the corresponding actions. In the following, we present the framework of the algorithm along with the underlying intuition. A table summarizing key concepts and variables used in the algorithm is provided in Table 1. The main structure of the algorithm is given in Algorithm 2, and the pseudocode of the modules used is included in Appendix B.

**Decision Making**

During each time step $t$, the agent chooses an arm $a(t)$ from the belief arm set $\mathcal{B}(t)$[1]. The belief arm set is a collection of arms that is ready for either distribution exploration or cool-down exploration. The selection for $a(t)$ is governed by a modified Thompson Sampling algorithm: in most cases, the agent adheres to the standard Thompson Sampling rule, except when an arm is deemed "urgent" for cool-down exploration. This modification addresses scenarios where conventional Thompson Sampling might fail to adequately prioritize cool-down exploration for certain important arms (as defined in Section 3.1). Without sufficient cool-down exploration, the reward mean estimate for these arms may become unreliable, leading to suboptimal decisions. By incorporating this modification, the agent ensures that important arms receive the necessary exploration, balancing immediate reward optimization with long-term decision-making efficacy. Additional details about the bifurcated decision-making approach are provided in Section 6.3, and the full implementation of the decision-making process is described in Algorithm 2.

---

[1]if $\mathcal{B}(t) = \emptyset$, then the agent randomly selects an arm according to Algorithm 5.

Table 1: Summary of Critical Variables and Concepts

| Concept | Definition or Role in the Algorithms |
|---|---|
| $\alpha^{(k)}(t), \beta^{(k)}(t)$ | Parameters for agent's estimate of success rate (beta distribution) |
| $\mathcal{B}(t)$ | The belief arm set, which stands for the collection of arms that are ready for distribution exploration or cool-down exploration at time $t$ (check more information in $CD\_Explore^{(k)}(t)$). |
| $\mathcal{B}'(t)$ | A collection of arms that are at the end of the cool-down period in the agent's belief at time $t$. |
| $CD\_Explore^{(k)}(t)$ | A variable used for estimating the active state of arm $k$. When $CD\_Explore^{(k)}(t) = \infty$, arm $k$ is ensured to be active and ready for distribution exploration; when $CD\_Explore^{(k)}(t) \leq 0$, arm $k$ is ready for cool-down exploration. |
| $check^{(k)}(t)$ | The accumulated count of times the agent engages in cool-down exploration for arm $k$ without attaining a 1-reward in $(c = L_u^{(k)}(t))$-run. This variable serves to gauge the agent's confidence in $L_u^{(k)}(t)$. |
| $c$-run | The collection of time instances $t$ satisfying $L_u^{(k)}(t) = c$. |
| $\mathcal{I}$ | Important arm set, a collection of arms that are the optimal active arms with high frequency, see Section 3.1. |
| $last\_CD^{(k)}(t)$ | The last time arm $k$ is selected for cool-down exploration before time $t$. |
| $l^{(k)}$ | The actual cool-down duration of arm $k$. |
| $L^{(k)}$ | A variable characterizes the sufficiency of cool-down exploration: if $L_u^{(k)}(t) \leq L^{(k)}$, then the cool-down exploration is considered as sufficient. The definition is in Section 3.2. |
| $L_u^{(k)}(t)$ | The estimated cool-down duration of arm $k$ at time $t$. |
| $L_{test}^{(k)}(t)$ | An intermediate variable used for testing if $L_u^{(k)}(t)$ reaches the actual cool-down duration $l^{(k)}$, which is designed to satisfy $L_{test}^{(k)}(t) \leq L_u^{(k)}(t)$, with the equality holds only when the agent has enough confidence to believe $L_u^{(k)}(t) = l^{(k)}$. |
| $p^{(k)}$ | The actual reward mean / success rate of arm $k$. |
| $\tilde{p}^{(k)}(t)$ | The estimated success rate of arm $k$ at time $t$, defined by $\tilde{p}^{(k)}(t) = \frac{\alpha^{(k)}(t)}{\alpha^{(k)}(t)+\beta^{(k)}(t)}$. |
| $waitlist^{(k)}(t)$ | A signal indicating whether the cool-down exploration of arm $k$ is suspended. Specifically, $waitlist^{(k)}(t) = 1$ signifies cool-down exploration suspension at time $t$ for arm $k$, and 0 otherwise. |

**Distribution Exploration**

During distribution exploration, the agent takes a single sample from arm $a(t)$ and updates $\alpha^{(a(t))}(t)$ and $\beta^{(a(t))}(t)$ according to the following equations:

$$\alpha^{(k)}(t+1) = \alpha^{(k)}(t) + X^{(k)}(t+1),$$
$$\beta^{(k)}(t+1) = \beta^{(k)}(t) + 1 - X^{(k)}(t+1).$$

The distribution exploration process is given in Algorithm 3 in Appendix B.

**Cool-Down Exploration**

The cool-down exploration for each arm $k$ comprises two stages: the quick jump stage and the confirmation stage, which is decided by comparing the estimated cool-down duration $L_u^{(k)}(t)$ and an intermediate testing value $L_{test}^{(k)}(t)$. The quick jump stage occurs when $L_{test}^{(k)}(t) < L_u^{(k)}(t) - 1$. During this phase, the agent utilizes a bisection algorithm to promptly update the estimated cool-down duration $L_u^{(k)}(t)$ based on sampling

---

**Algorithm 2:** Unknown Cool-Down Durations

---

**Input:** total time $T$, upper bound $D$

1  **Initialization** For $k = 1, \ldots, M$, set $\alpha^{(k)} = \beta^{(k)} = 1$, $L_u^{(k)} = D$, $L_{test}^{(k)} = \min(D-1, \lceil \frac{D}{2} \rceil)$, $\tilde{L}_{test}^{(k)} = 0$,
   $CD\_Explore^{(k)} = \infty$, $check^{(k)} = 0$, $waitlist^{(k)} = 0$, $last\_CD^{(k)} = 0$. Set $t = 0$.

2  **while** $t < T$ **do**

3     $t = t + 1$

4     **for** $k = 1, \ldots, M$ **do**

5        | The agent samples $\theta^{(k)}$ from $\text{Beta}(\alpha^{(k)}, \beta^{(k)})$

6     **end**

7     Let $\mathcal{B}$ be the set of arm $k$ satisfying $CD\_Explore^{(k)} \leq 0$ or $CD\_Explore^{(k)} = \infty$    `// those arms that`
       `are ready for cool-down exploration or distribution exploration`

8     **if** $\mathcal{B} \neq \emptyset$ **then**

9        Let $\mathcal{B}'$ be collection of arm $k$ in $\mathcal{B}$ satisfying $CD\_Explore^{(k)} = L_{test}^{(k)} - L_u^{(k)} + 1$

10       **if** $\mathcal{B}' \neq \emptyset$ and $\min_{k \in \mathcal{B}'} last\_CD^{(k)} < last\_CD^{(\arg\max_{k \in \mathcal{B}} \theta^{(k)})}$      `// urgent cool-down exploration`
       `requirement`

11       **then**

12        | $a(t) = \arg\min_{k \in \mathcal{B}'} last\_CD^{(k)}$      `// urgent cool-down exploration`

13       **else**

14        | $a(t) = \arg\max_{k \in \mathcal{B}} \theta^{(k)}$      `// Thompson Sampling rule`

15       **end**

16       **if** $CD\_Explore^{(a(t))} = \infty$ **then**

17        | Distribution Exploration

18       **else**

19        | $last\_CD^{(a(t))} = t$

20        | CD exploration

21       **end**

22     **else**

23       | Random Sample      `// if no arm is ready for either exploration, then randomly select an arm`

24     **end**

25     Post-Update

26 **end**

---

outcomes. The confirmation stage arises when $L_{test}^{(k)}(t) = L_u^{(k)}(t) - 1$. Here, the agent carefully refines its confidence in $L_u^{(k)}(t)$. The cool-down exploration process is delineated in Algorithm 4 in Appendix B.

**Suspending or Restarting of Cool-Down Exploration**

To avoid insufficient and excessive cool-down exploration simultaneously, a dynamic criterion is employed to restart, suspend, or terminate the cool-down exploration. This criterion measures agent's confidence on the estimated cool-down duration by comparing $(1 - \tilde{p}^{(k)}(t))^{check^{(k)}(t)}$ with $\frac{1}{t \log t}$, where $\tilde{p}^{(k)}(t)$ and $check^{(k)}(t)$ are defined in Table 1, and based on which, helps the agent to decide whether to suspend or restart the cool-down exploration for arm $k$. By doing so, we ensure a very low probability of insufficient cool-down exploration, see the analysis in Appendix A.4.2, while preventing excessive exploration, as detailed in Lemma 8. Further details and variable updates are provided in Algorithm 6.

## 3   Additional Concepts

### 3.1   Important Arms

To delineate the important arm set, let us start by examining a simplified scenario where the reward distribution for each arm $k$ is a constant $p^{(k)}$, rather than a Bernoulli distribution, while maintaining all other settings. The agent possesses knowledge of both $p^{(k)}$ and $l^{(k)}$ for each arm $k = 1, \ldots, M$. In this setup,

the optimal policy becomes deterministic: the agent consistently selects $\arg\max_{k \in \mathcal{A}(t)} p^{(k)}$. Additionally, it is evident that the chosen arm follows a periodic pattern given that all arms are initialized to be active at $t = 0$. Let $\mathcal{P}$ denote the repetitive sequence of labels corresponding to the arms chosen in each period, the important arm set $\mathcal{I}$ is then defined as the non-repetitive copy of $\mathcal{P}$. Return to the bandit setting studied in this paper, the following lemma shows the property of $\mathcal{I}$ defined as above.

**Lemma 1.** *If the bandit algorithm satisfies that for every $a > 0$, the number of sub-optimal selects is $o(t^a)$, then for any arm $i \in \mathcal{I}$ and $w \in (0, 1)$, $\mathbf{P}(|\mathcal{T}_i^o(t)| < t^w) \leq O(e^{-t})$. For any arm $j \notin \mathcal{I}$ and $w \in (0, 1)$, $\mathbf{P}(|\mathcal{T}_j^o(t)| > t^w) \leq O(e^{-t^{\frac{w}{2}}})$.*

The above lemma shows that the frequency of an important arm being the optimal active arm is high while that of the other arms is extremely low.

### 3.2 "Sufficient" Cool-Down Exploration

We mentioned in Section 1.3 that insufficient cool-down exporation is likely to result in a linear regret. To quantify the "sufficiency" of the cool-down exploration, we define $L^{(k)}$ as the largest value such that $\mathcal{I}$ remains unchanged if the cool-down duration of arm $k$ changes from $l^{(k)}$ to $L^{(k)}$. In this sense, for $k \notin \mathcal{I}$, $L^{(k)} = D$. We say the cool-down exploration for arm $k$ is sufficient for arm $k$ at time $t$ if and only if $L_u^{(k)}(t) \leq L^{(k)}$. Note that for each arm $k = 1, \ldots, M$, $L_u^{(k)}(t)$ may still be updated after it reaches $L^{(k)}$.

## 4 Theoretical Results and Comparisons

### 4.1 Known Cool-Down Durations

In the scenario where the cool-down duration $l^{(k)}$ is known to the agent for each arm $k$, we present the following lower bound result and an asymptotically optimal upper bound result using Algorithm 1.

**Theorem 1** (Lower Bound). *If the agent is aware of $l^{(k)}$ for each arm $k = 1, \ldots, M$, then the lower bound of the regret satisfies*

$$R(T) \sim \sum_{j=2}^{M} \frac{p^{(\min(j-1,|\mathcal{I}|))} - p^{(j)}}{K(p^{(j)}, p^{(\min(j-1,|\mathcal{I}|))})} \log T \quad as\ T \to \infty,$$

*where $K(p, q)$ denotes the Kullback-Leibler divergence*

$$K(p, q) = p \log \frac{p}{q} + (1 - p) \log \frac{1 - p}{1 - q}.$$

**Theorem 2** (Upper Bound). *If the agent is aware of the cool-down duration of each arm, then under Algorithm 1, for any $\epsilon$, there exist a constant $C(\epsilon) = O(\frac{1}{\epsilon^2})$, such that the regret satisfies that*

$$R(T) \leq (1 + \epsilon) \sum_{j=2}^{M} \sum_{i=1}^{j-1} \frac{(p^{(i)} - p^{(i+1)}) \log T}{K(p^{(j)}, p^{(\min(i,|\mathcal{I}|))})} + C(\epsilon).$$

**Remark 1** (Comparison with conventional bandit results). *The classic bandit problem without considering the cool-down durations can be regarded as a special instance within our framework, wherein each arm $k = 1, \ldots, M$ satisfies $l^{(k)} = 0$. (Lai et al., 1985) provides an asymptotic lower bound with order $\Omega(\sum_{k=2}^{M} \frac{p^{(1)} - p^{(k)}}{K(p^{(k)}, p^{(1)})} \log T)$, and (Agrawal & Goyal, 2017) provides a (near-optimal) asymptotic upper bound with order $O((1 + \epsilon) \sum_{k=2}^{M} \frac{p^{(1)} - p^{(k)}}{K(p^{(k)}, p^{(1)})} \log T)$ by using Thompson sampling. Note that $\mathcal{I} = \{1\}$ when $l^{(k)} = 0$ for each $k = 1, \ldots, M$, then the regret bound in Theorem 2 satisfies that*

$$R(T) \leq (1 + \epsilon) \sum_{j=2}^{M} \sum_{i=1}^{j-1} \frac{(p^{(i)} - p^{(i+1)}) \log T}{K(p^{(j)}, p^{(1)})} + C(\epsilon)$$

$$= (1 + \epsilon) \sum_{j=2}^{M} \frac{p^{(1)} - p^{(j)}}{K(p^{(j)}, p^{(1)})} + C(\epsilon).$$

*It is easy to see both our lower bound in Theorem 1 and the upper bound in Theorem 2 boil down to the classic results under the conventional bandit setting.*

The detailed analysis for Theorems 1 and 2 can be found in Appendix A.

### 4.2 Sleeping Bandit Scenario

In the general sleeping bandit framework addressed in (Kleinberg et al., 2010; Chatterjee et al., 2017), a periodic pattern of arm availability is not presupposed. The set $\mathcal{A}(t)$, representing the available arms at any given time $t$, may not adhere to any predictable pattern and is not known before $t$ for all $t > 0$. Nevertheless, it is assumed that $\mathcal{A}(t)$ is known to the agent at time $t$. Under these conditions, Algorithm 1 is still applicable to this scenario. Furthermore, by defining $\mathcal{I} = \{1, \ldots, M\}$, the analysis conducted in the proof of Theorem 2 can be directly transferred to this context. Thus, we are able to establish the following regret upper bound for the general sleeping bandit setting.

**Theorem 3.** *For the sleeping bandit scenario where the agent is aware of $\mathcal{A}(t)$ at each time instance $t$, then under Algorithm 1, for any $\epsilon > 0$, there exists a constant $\tilde{C}(\epsilon) = O(\frac{1}{\epsilon^2})$, such that the regret satisfies*

$$R(T) \leq (1 + \epsilon) \sum_{j=2}^{M} \sum_{i=1}^{j-1} \frac{(p^{(i)} - p^{(i+1)}) \log T}{K(p^{(j)}, p^{(i)})} + \tilde{C}(\epsilon).$$

**Remark 2** (Comparison with the sleeping bandit results). *From (Kleinberg et al., 2010; Chatterjee et al., 2017), the regret upper bound satisfies that*

$$\frac{R(T)}{\log T} \leq \sum_{j=2}^{M} \sum_{i=1}^{j-1} \frac{32(p^{(i)} - p^{(i+1)})}{p^{(i)} - p^{(j)}} + o(1).$$

*By using Pinsker's inequality, we can further bound the regret in Theorem 3 as*

$$\frac{R(T)}{\log T} \leq \sum_{j=2}^{M} \sum_{i=1}^{j-1} \frac{(1 + \epsilon)(p^{(i)} - p^{(i+1)})}{K(p^{(j)}, p^{(i)})} + o(1)$$

$$\leq \sum_{j=2}^{M} \sum_{i=1}^{j-1} \frac{(1 + \epsilon)(p^{(i)} - p^{(i+1)})}{2(p^{(i)} - p^{(j)})} + o(1).$$

*The regret bound we derive significantly surpasses those found in existing literature, primarily due to two key factors. Firstly, we employ the Thompson Sampling algorithm instead of UCB1, as utilized in (Kleinberg et al., 2010), leading to a superior bound. Secondly, in comparison to (Chatterjee et al., 2017), we integrate the near-optimal bound technique outlined in (Agrawal & Goyal, 2017) into the time-varying scenario, resulting in an enhanced upper bound for the number of sub-optimal selections. We have also implemented numerical comparison, see Figure 1.*

### 4.3 Unknown Cool-Down Durations

**Theorem 4** (Upper Bound). *For any $\epsilon > 0$, there exist constant $\hat{C}_1(\epsilon) = O(1)$ and $\hat{C}_2(\epsilon) = O(\frac{-\log \epsilon}{\epsilon^2})$, such that under Algorithm 2, the regret satisfies*

$$R(T) \leq (1 + \epsilon) \min \Bigg\{ \sum_{j=2}^{M} \bigg( \sum_{i=1}^{j-1} \bigg( \frac{2(p^{(i)} - p^{(i+1)})}{K(p^{(j)}, p^{(\min(i, |\mathcal{I}|))})} + 1 \bigg) + 1 \bigg),$$

$$\sum_{j=2}^{M} \sum_{i=1}^{j-1} \frac{(p^{(i)} - p^{(i+1)})}{K(p^{(j)}, p^{(\min(i, |\mathcal{I}|))})} + \sum_{j=2}^{M} \frac{(1 + \epsilon)p^{(i)}}{-\log(1 - p^{(j)})} \Bigg\} \log T$$

$$+ \hat{C}_1(\epsilon) \log \log T + \hat{C}_2(\epsilon),$$

**Remark 3** (Comparison with known cool-down duration setting). *The additional parts compared with the regret bound in Theorem 2 is mostly due to the additional efforts the agent makes on learning the cool-down duration of each arm, as is analyzed in the proof of Theorem 4. The comparison of the practical performance of the agent under both settings we consider is given in Figure 2a.*

In below, we provide the sketched proof of Theorem 4, the complete proof can be found in Appendix A.

**Sketched proof of Theorem 4:** In the proof of Theorem 4, we identify four situations where a loss in reward may occur: the first situation arises from insufficient distribution exploration, the second from insufficient cool-down exploration, and the third and fourth from excessive cool-down explorations. To quantify the error in the first situation, we utilize Corollaries 1 and 2, extending the conventional uncertainty analysis associated with Thompson Sampling to our problem setting. For the second situation, we show in Lemmas 6 and 7 that the probability of excluding the optimal arm from $\mathcal{B}(t)$ due to insufficient cool-down exploration diminishes over time under Algorithm 2. The assessments for the third and fourth situations employ both the analysis in Lemma 6 and Lemma 8, with the latter one establishing an upper bound on unnecessary cool-down explorations once the agent's estimated cool-down duration for each arm reaches the true value. Note that the proof of Lemma 8 necessitates an assessment of the accuracy of $\tilde{p}^{(k)}(t)$ for each arm $k$ and time $t$, which is provided in Lemma 5.

## 5 Numerical Evaluations

In this section, we evaluate the performance of our proposed algorithms. We consider a 10-armed bandit problem, each arm follows a Bernoulli distribution with distinct success rate. For all the experiments in this section, we set the cool-down duration upper bound $D$ as 10. The total time $T$ is set to be 100000 and we provide the averaged performance over 50 runs with the corresponding error bar.

This paper is the first to address the bandit scenario in which the agent is unaware of whether each arm is active, referred to as the unknown cool-down duration scenario. Consequently, no direct baseline comparison is available for this setting. Instead, we benchmark our approach against existing methods from the sleeping bandit literature. The result is presented in Figure 1. Furthermore, we evaluate the agent's performance when the cool-down durations are known (Algorithm 1) versus unknown (Algorithm 2), to quantify the performance degradation caused by the lack of this knowledge. We also investigate the special case where $l^{(k)} = 0$ for all $k = 1, \ldots, M$, corresponding to the classic bandit framework. In this case, Algorithm 1 aligns with the conventional Thompson Sampling algorithm. The performance results for these scenarios are provided in Figures 2a and 2b.

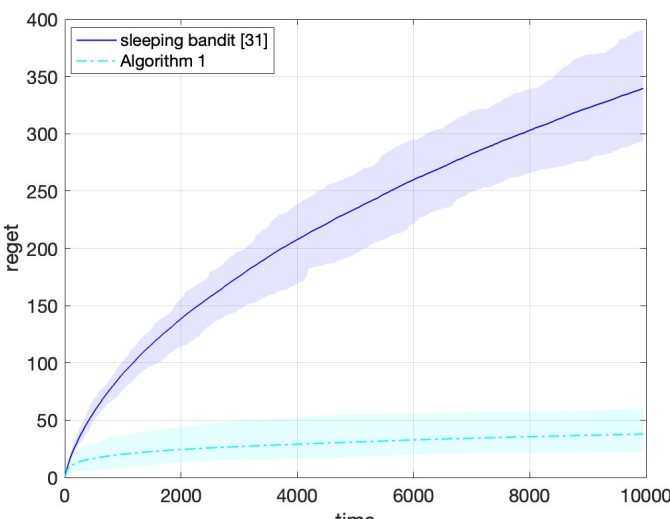

Figure 1: Numerical comparison for Algorithm 1 and that in (Chatterjee et al., 2017) in sleeping bandit senario

Figure 1 clearly illustrates the superiority of our algorithm compared to existing method in the sleeping bandit scenario, aligning with the theoretical comparison discussed in Remark 2. Examination of Figures 2a and 2b reveals that in both scenarios studied in this paper, the regret trends logarithmically. This observation

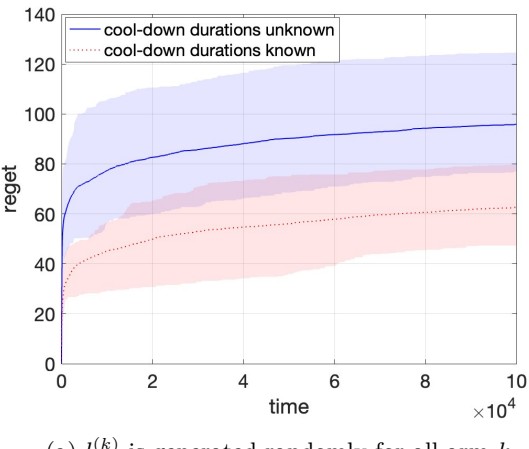
(a) $l^{(k)}$ is generated randomly for all arm $k$

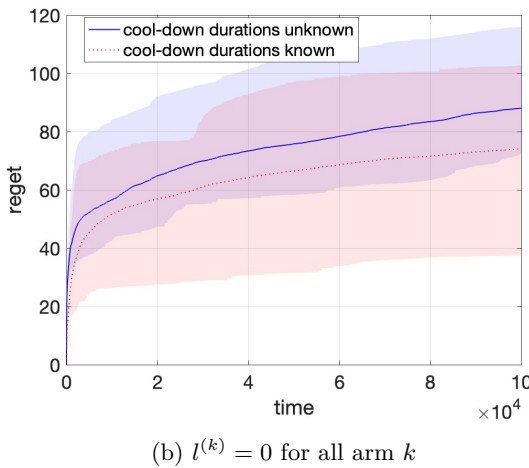
(b) $l^{(k)} = 0$ for all arm $k$

Figure 2: Numerical comparison for Algorithm 1 and Algorithm 2

aligns with the theoretical results presented in Theorem 2 and Theorem 4. Moreover, despite the integration of an additional cool-down exploration process and the complexity added by the decision-making bifurcation, the regret incurred when cool-down durations are unknown does not significantly exceed that observed when the cool-down durations are known. This demonstrates the effectiveness and advantage of our specifically tailored cool-down exploration and decision-making processes.

## 6 Further Discussions and Simulations

### 6.1 More General Reward Setting

Although this paper focuses on the Bernoulli reward distribution for simplicity, both Algorithm 1 and Algorithm 2 can be readily extended to more general stochastic reward settings. This follows from the fact that both algorithms employ the Thompson sampling principle, which naturally generalizes to arbitrary stochastic rewards (see Algorithm 2 in (Agrawal & Goyal, 2012)). With corresponding modifications in the analysis, our results can thus be extended to general stochastic reward distributions.

### 6.2 Naive Thompson Sampling

As mentioned in Section 1.3, employing conventional Thompson Sampling without any cool-down exploration process typically results in linear regret in scenarios where each arm has a nonzero cool-down duration. This is because doing so would result in the agent incorporating the 0-reward obtained due to selecting an inactive arm into its success rate estimation, which biases the estimate, and consequently may cause the regret to escalate linearly. To better illustrate this, we conduct the following simulation by considering a 10-armed bandit problem, with each arm following a Bernoulli distribution characterized by distinct success rates. The total time $T$ is set to 10000. Figure 3 depicts the averaged curve over 50 runs.

### 6.3 Bifurcation in Decision-Making

For scenarios where the agent is unaware of the cool-down duration of each arm, we introduce a bifurcated decision-making in Algorithm 2, i.e., lines 10-15 in Algorithm 2. As discussed in **Decision Making** in Section 2.2, the agent chooses between the optimal arm in the agent's belief decided by Thompson Sampling rule (line 14 in Algorithm 2) and the one that is the most urgent for cool-down exploration (line 12 in Algorithm 2). Whether an arm is deemed as "urgent" for cool-down exploration is decided by two factors: (1) whether it is at the end of the current cool-down period (2) the last time it is selected for cool-down exploration (see line 10 in Algorithm 2). Such a decision-making criteria sometimes prompts the agent to choose an arm other than the optimal arm in the agent's belief, as otherwise, the cool-down duration of some important arms may never be sufficiently explored. We will use an example to further explain this. Suppose at time $t$, there exists a sub-optimal arm $k$ satisfying $l^{(k)} < L_u^{(k)}(t) = L_{test}^{(k)}(t) + 1$. For such a situation, whenever arm $k$ becomes inactive, the agent has only one opportunity to initiate cool-down exploration on

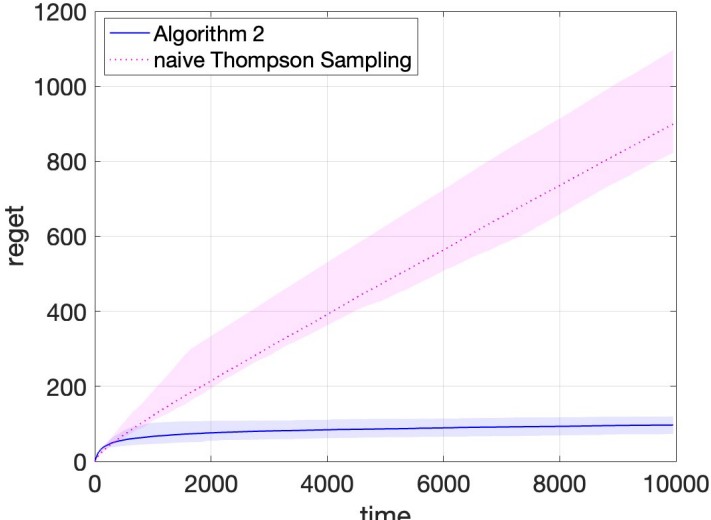

Figure 3: Simulation results for Algorithm 2 and naive Thompson Sampling when cool-down durations are unknown

it. If, at that moment, there exists a "superior" arm with a higher estimated success rate in the agent's belief arm set, without adding the decision-making bifurcation, the agent is likely to choose the superior arm based on the Thompson Sampling rule instead of exploring the cool-down duration of arm $k$. If this situation recurs frequently, the agent may never acquire sufficient exploration of the cool-down duration of arm $k$. We conduct the following simulation to better elucidate the necessity of the bifurcated decision-making process.

Consider the following example, there are three arms with success rates 0.9, 0.8, 0.5 and cool-down durations 2, 0, 0 respectively. Let $D = 2$. Since $l^{(2)} = 0$, arm 2 is always active. Let us assume the agent fails to obtain reward 1 during the first 5 cool-down explorations of arm 2, which happens with probability at least $0.2^{10}$. Given $T = 10000$, the performance results of the agent under Algorithm 2 and that without the decision-making bifurcation are illustrated in Figure 4.

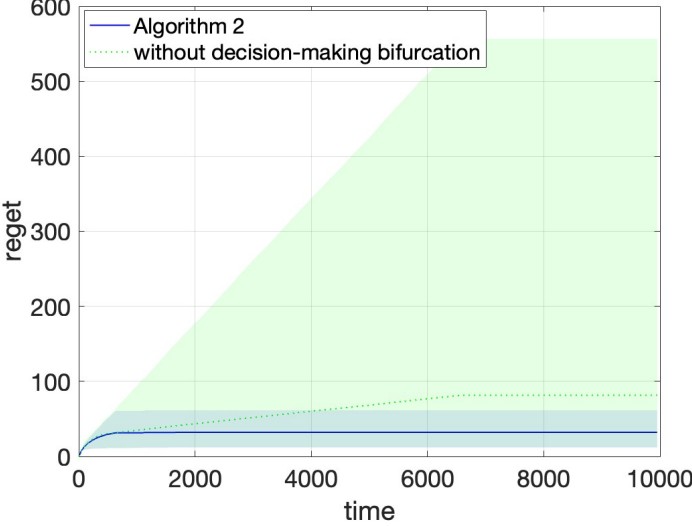

Figure 4: Simulation results for Algorithm 2 and the algorithm without decision-making bifurcation

In Figure 4, it is evident that the worst performance, along with the variance of regret, is significant. Since the probability of the situation in the simulation is a constant-large value, it's plausible that the agent may experience linear regret without the designed decision-making bifurcation.

### 6.4 Non-Stop Cool-Down Exploration

As detailed in Section 1.3, addressing the problem when the cool-down durations are unknown necessitates a policy for the agent to strategically pause and resume the cool-down exploration for each arm, thereby preventing excessive exploration of the cool-down duration. This process is delineated in Lines 3-20 of Algorithm 6. In the subsequent simulation, we explore the ramifications of not implementing such a process. We examine a 10-armed bandit problem, where each arm follows a Bernoulli distribution characterized by distinct success rates. The total time $T$ is set to 10000, with a total of 50 runs. The simulation results are presented in Figure 5.

It can be clearly observed from Figure 5 that in the absence of criteria for pausing or resuming cool-down exploration, the agent experiences linear regret, despite accurately estimating both the success rate and cool-down duration for each arm. This observation aligns with the explanation provided in Section 1.3.

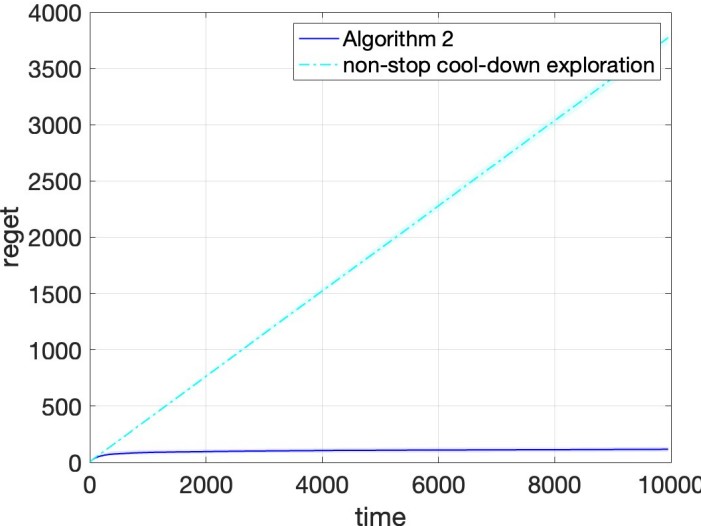

Figure 5: Simulation results for Algorithm 2 and the one without stopping policy

## 7 Conclusion

We explore a variant of dynamic bandits in which each arm's availability follows a periodic pattern. We have developed modified Thompson Sampling algorithms tailored for scenarios where the agent is either aware or unaware of the cool-down duration of each arm. Both adaptations are demonstrated to achieve logarithmic regret. Notably, the algorithm designed for the unknown cool-down duration scenario is the first to tackle the bandit problem where the active state of each arm is unknown. Furthermore, in the case where the cool-down duration is known—particularly in the sleeping bandit setting—our algorithm demonstrates superior performance compared to existing approaches (Kleinberg et al., 2010; Chatterjee et al., 2017). Potential directions for future work include exploring dependencies among different arms and conducting a detailed evaluation of empirical regret.

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
