## A    Regret Analysis

### A.1    Preliminaries

**Lemma 2** (Hoeffding's inequality (Hoeffding, 1963))**.** *For a Bernoulli trail with success rate p, let $S_n$ be the total reward of n trails, then*

$$\mathbf{P}(S_n \geq np + a) \leq \exp\left(\frac{-2a^2}{n}\right)$$

$$\mathbf{P}(S_n \leq np - a) \leq \exp\left(\frac{-2a^2}{n}\right)$$

**Lemma 3** (Thompson Sampling near-optimal regret (Agrawal & Goyal, 2017))**.** *For standard bandit problem without considering cool-down periods, under Thompson Sampling, for any $k = 2, \ldots, M$ and $\epsilon > 0$, let $L_k(T) = \frac{(1+\epsilon)\log T}{K(p^{(k)}, p^{(1)})}$, there exists a constant $c(\epsilon) = O(\frac{1}{\epsilon^2})$, such that*

$$\sum_{t=1}^{T} \mathbf{P}(a(t) = k, n_k(t) > L_k(T)) \leq c(\epsilon).$$

*Proof.* The lemma is a direct consequence of (Agrawal & Goyal, 2017, Lemmas 2.10, 2.11) and the analysis in (Agrawal & Goyal, 2017, Lemma 2.12) by setting $x_k \in (p^{(k)}, p^{(1)})$ in the statement such that $K(x_k, p^{(1)}) = \frac{K(p^{(k)}, p^{(1)})}{1+\epsilon}$ and $y_k \in (x_k, p^{(1)})$ in the statement such that $K(x_k, y_k) = \frac{K(p^{(k)}, p^{(1)})}{(1+\epsilon)^2}$. ∎

### A.2    Notations

For any $k \in \{1, \ldots, M\}$, let $\mathcal{T}_k^a(t) = \{\tau \leq t : k \in \mathcal{A}(t)\}$ be the collection of time instances such that arm $k$ is active, let $\mathcal{T}_k^o(t) = \{\tau \leq t : k = \arg\max_{i \in \mathcal{A}(\tau)} p^{(i)}\}$ be the collection of time instance $\tau \leq t$ such that arm $k$ is the actual optimal active arm at time $t$. Let $\mathcal{C}_1^{(k)}(t)$ be the sequence of the time instances when the agent does distribution exploration on arm $k$ till time $t$, $\mathcal{C}_2^{(k)}(t)$ be the sequence of the starting time of each CD exploration on arm $k$ till time $t$ and $\tilde{\mathcal{C}}_2^{(k)}(t)$ be the sequence of the time instances when the agent enters the confirmation step of arm $k$. Let $m^{(k)}(t)$ be the number of samplings in CD exploration of arm $k$ till time $t$, we have $|\mathcal{C}_2^{(k)}(t)| \leq m^{(k)}(t)$. Moreover, from Lemma 4, it holds that

$$|\tilde{\mathcal{C}}_2^{(k)}(t)| \geq m^{(k)}(t) - c_2^{(k)}. \tag{1}$$

For any $j = 2, \ldots, M$ and $i = 1, \ldots, j-1$, let $\mathcal{T}_{i,j}(T)$ be the collection of time instances over total time $T$ such that both arm $i$ and $j$ are active, and $\mathcal{T}_{1:i,j}(T)$ be the collection of time instances over total time $T$ such that at least one of arms in $\{1, \ldots, i\}$ is active, and arm $j$ is active. Let $N_{i,j}(T), M_{i,j}(T)$ be the size of the collection of time $t$ over total time $T$ such that $a(t) = j$, $a^*(t) \in \{1, \ldots, i\}$ and the agent enters the "urgent" cool-down exploration (line 12 in Algorithm 2) for arm $j$ respectively, and $\tilde{N}_{i,j}(T), \tilde{M}_{i,j}(T)$ be the size of the collection of time $t$ over total time $T$ such that $a(t) = j$, $a^*(t) = i$ and the agent selects arm $j$ following the Thompson sampling rule (line 14 i Algorithm 2) for distribution exploration and cool-down exploration respectively. Specifically, when the cool-down durations are known to the agent, $N_{i,j}(T)$ (resp. $\tilde{N}_{i,j}(T)$) simply refers to the accumulated number of times over $T$ that the agent selects arm $j$ when the optimal arm is in $\{1, \ldots, i\}$ (resp. is $\{i\}$).

### A.3    Known Cool-Down Duration

We start with the proof of Lemma 1, which provides a property of the important arm set defined in Section 3.1.

*Proof of Lemma 1:* Let $S(t)$ be the stack of $S_k(t)$, which stands for the remaining cool-down time for each arm $k = 1, \ldots, M$ to become active. When $k \in \mathcal{A}(t)$, $S_k(t) = 0$. Then $S(t) = 0$ represents for the state that all the arms are active at time $t$. From the definition of $\mathcal{I}$, if $S(t) = 0$, then for any $i \in \mathcal{I}$, with probability at least $p_1 \triangleq \prod_{k \in \mathcal{P}} p^{(k)}$, $i = a^*(\tau)$ for some $\tau = t, \ldots, t + |\mathcal{P}| - 1$, where $\mathcal{P}$ is defined in Section 3.1. Besides, for any state $S(t)$, if all the trial fail starting from $t$, then there exists a $\tau \in [t, t + D]$, such that $S(\tau) = 0$. This

implies that for any $S(t)$, with probability at least $p_2 \triangleq \prod_{i \in \mathcal{I}} p^{(i)}(p^{(M)})^{D-|\mathcal{I}|}$, the state becomes 0 within $D$ steps. Combining the above results yields that for any $t \leq T - D - |\mathcal{P}| + 1$ and $i \in \mathcal{I}$, with probability at least $p_1 p_2$, there exists a $\tau \in [t, t + D + |\mathcal{P}| - 1]$, such that $a^*(t) = i$. This implies that for any $i \in \mathcal{I}$ and integer $c$, it holds that

$$\mathbf{P}(|\mathcal{T}_i^o(t)| < c) \leq F(c), \tag{2}$$

where $F$ stands for the cumulative distribution function of $B(\lfloor \frac{T}{D+|\mathcal{P}|} \rfloor, p_1 p_2)$, here $B(n, p)$ represents for the binomial distribution of the number of successful trials over $n$ trials with success rate $p$. For any $n, p$ in the domain, it holds for all $w \in (0, 1)$ that

$$F(n^w) = \sum_{k=0}^{n^w} \binom{n}{k} p^k (1-p)^{n-k}$$
$$\leq n^w n^{n^w} (1-p)^{n-n^w}.$$

Then $\log F(n^w) \leq (w + n^w) \log n + (n - n^w) \log(1 - p) = O(-n)$, and therefore $F(n^w) \leq O(e^{-n})$. Setting $n = \lfloor \frac{T}{D+|\mathcal{P}|} \rfloor$, $p = p_1 p_2$ and $c = t^w$ and substituting the above result to equation 2 yield that for any $i \in \mathcal{I}$ and $w \in (0, 1)$, there holds

$$\mathbf{P}(|\mathcal{T}_i^o(t)| < t^w) \leq O(e^{-w}),$$

which completes the first part of the proof.

From the definition of $\mathcal{I}$, for any $j \notin \mathcal{I}$, the only chance that $j = a^*(t)$ is when the agent selects a sub-optimal arm at some time $t' < t$ and $S(\tau) \neq 0$ for all $\tau = t' + 1, \ldots, t$. Let $n_s(t)$ denote the total number of sub-optimal selects over time $t$. Using the Pigeonhole principle, for any $j \notin \mathcal{I}$ and $w \in (0, 1)$, if $|\mathcal{T}_j^o(t)| > t^w$, there exists a $t_1 < T$, such that the agent selects a sub-optimal arm at time $t_1$, and $S(\tau) \neq 0$ for all $\tau = t_1 + 1, \ldots, t_1 + \lceil \frac{t^w}{n_s} \rceil$, then

$$\mathbf{P}(|\mathcal{T}_j^o(t)| > t^w) \leq (1 - p_2)^{\lceil \frac{t^w}{n_s(t)} \rceil}.$$

Set $a = \frac{w}{2}$ in the lemma statement, then there exists a $c_w$, such that $n_s(t) \leq c_w t^{\frac{w}{2}}$. Then

$$\mathbf{P}(|\mathcal{T}_j^o(t)| > t^w) \leq (1 - p_2)^{\lceil \frac{t^{\frac{w}{2}}}{c_w} \rceil} \leq \left( (1 - p_2)^{\frac{1}{c_w}} \right)^{t^{\frac{w}{2}}} = O(e^{-t^{\frac{w}{2}}}),$$

which completes the proof. ∎

Then we are able to prove Theorem 1 and Theorem 2.

*Proof of Theorem 1:* For any arm $j = 2, \ldots, M$ and $t > 0$, similar to the analysis in (Lai et al., 1985, Theorem 2), for any $\epsilon > 0$, it holds that

$$\lim_{t \to \infty} \mathbf{P}\left( N_{j-1,j}(t) \geq \frac{(1 - \epsilon) \log t}{K(p^{(j)}, p^{(a^*(t))})} \right) = 1.$$

For any $i \in \mathcal{I}$, let $T_i = \arg\max_{t \leq T} a^*(t) = i$, then for any $j = 2, \ldots, M$ and $i = 1, \ldots, \min(j - 1, |\mathcal{I}|)$, there holds

$$\lim_{T \to \infty} \mathbf{P}\left( N_{j-1,j}(T) \geq \frac{(1 - \epsilon) \log T_i}{K(p^{(j)}, p^{(i)})} \right) = 1.$$

Combining the above inequality with Lemma 1, for any $\epsilon > 0, w \in (0, 1), j = 2, \ldots, M$ and $i = 1, \ldots, \min(j - 1, |\mathcal{I}|)$, it holds that

$$\lim_{T \to \infty} \mathbf{P}\left( N_{j-1,j}(T) \geq \frac{w(1 - \epsilon) \log T}{K(p^{(j)}, p^{(i)})} \right) = 1,$$

which implies that for any $j = 2, \ldots, M$ and $i = 1, \ldots, \min(j - 1, |\mathcal{I}|)$, it holds that

$$\liminf_{T \to \infty} \frac{\mathbf{E}(N_{j-1,j}(T))}{\log T} \geq \frac{1}{K(p^{(j)}, p^{(i)})}.$$

From Lemma 1, for any $i \notin \mathcal{I}$, it holds that

$$\liminf_{T \to \infty} \frac{\mathbf{E}(\tilde{N}_{i,j}(T))}{\log T} = o(1),$$

where we also make use of the property that $|\mathcal{T}_i^o| \geq \tilde{N}_{i,j}(T)$. Then for any $j = 2, \ldots, M$, it holds that

$$\liminf_{T \to \infty} \frac{\mathbf{E}(N_{\min(j-1,|\mathcal{I}|),j}(T))}{\log T} \geq \frac{1}{K(p^{(j)}, p^{(i)})}.$$

Since $K(p, q)$ is increasing in terms of $q$ for $q > p$, for any $j = 2, \ldots, M$, it holds that

$$\liminf_{T \to \infty} \frac{\mathbf{E}(N_{\min(j-1,|\mathcal{I}|),j}(T))}{\log T} \geq \frac{1}{K(p^{(j)}, p^{(\min(j-1,|\mathcal{I}|))})}. \tag{3}$$

The regret $R(T)$ satisfies that

$$R(T) \geq \sum_{j=2}^{M} \mathbf{E}(N_{\min(j-1,|\mathcal{I}|),j}(T))(p^{(\min(j-1,|\mathcal{I}|))} - p^{(j)}).$$

Substituting equation 3 to the above inequality, we obtain that

$$\liminf_{T \to \infty} \frac{R(T)}{\log T} \geq \sum_{j=2}^{M} \frac{p^{(\min(j-1,|\mathcal{I}|))} - p^{(j)}}{K(p^{(j)}, p^{(\min(j-1,|\mathcal{I}|))})},$$

which completes the proof. ∎

*Proof of Theorem 2:* For any $j = 2, \ldots, M$ and $i = 1, \ldots, j - 1$, let

$$L_{i,j}(T) = \frac{(1 + \epsilon) \log(|\mathcal{T}_j^a(T) \cap \mathcal{T}_i^o(T)|)}{K(p^{(j)}, p^{(i)})}.$$

Similar to the analysis of Lemma 3, for any $\epsilon > 0$, $j = 2, \ldots, M$ and $i = 1, \ldots, j - 1$, it holds that

$$\mathbf{P}\left(\sum_{t \in \mathcal{T}_j^a(T) \cap \mathcal{T}_i^o(T)} \mathbf{1}(a(t) = j, n_j(t) > L_{i,j}(T))\right) \leq c(\epsilon).$$

From Lemma 1, for $i \in \mathcal{I}$ and $j = i + 1, \ldots, M$, it holds that

$$L_{i,j}(T) \leq \frac{(1 + \epsilon) \log T}{K(p^{(j)}, p^{(i)})},$$

and for $i \notin \mathcal{I}$, and $j = i + 1, \ldots, M$, it holds that

$$L_{i,j}(T) \leq \frac{(1 + \epsilon) \log(|\mathcal{T}_i^o(T)|)}{K(p^{(j)}, p^{(i)})} = o(\log T).$$

Then for any $j = 2, \ldots, M$, $i = 1, \ldots, j - 1$, and $\epsilon > 0$, there exists a constant $c_1(\epsilon) = O(1)$, such that when $T \geq c_1(\epsilon)$, for any $j = 2, \ldots, M$ and $i = 1, \ldots, j - 1$, it holds that

$$L_{i,j}(T) \leq \frac{(1 + \epsilon) \log T}{K(p^{(j)}, p^{(\min(i,|\mathcal{I}|))})} \triangleq \tilde{L}_{i,j}(T).$$

Note that

$$\mathbf{E}(N_{i,j}(T)) \leq \mathbf{E}\Big(N_{i,j}(T), n_j(T) \leq \max_{i<j} L_{i,j}(T)\Big) + \mathbf{E}\Big(N_{i,j}(T), n_j(T) > \max_{i<j} L_{i,j}(T)\Big).$$

For any $j = 2, \ldots, M$ and $i = 1, \ldots, j-1$, $N_{i,j}(T) \leq n_j(T)$ from definition, the former term is upper bounded by $\max_{i<j} L_{i,j}(T) \leq \max(\tilde{L}_{i,j}(T), c_1(\epsilon))$, and the latter term satisfies

$$
\begin{aligned}
\mathbf{E}\Big(N_{i,j}(T), n_j(T) > \max_{i<j} L_{i,j}(T)\Big) &\leq \mathbf{E}\Big(N_{i,j}(T), n_j(T) > L_{i,j}(T)\Big) \\
&= \mathbf{P}\Big(\sum_{t \in (\cup_{k=1}^{i} \mathcal{T}_k^o) \cap \mathcal{T}_j^a} \mathbf{1}\Big(a(t) = j, n_j(T) > L_{i,j}(T)\Big)\Big) \\
&= \mathbf{P}\Big(\sum_{k=1}^{i} \sum_{t \in \mathcal{T}_j^a(T) \cap \mathcal{T}_i^o(T)} \mathbf{1}\Big(a(t) = j, n_j(T) > L_{i,j}(T)\Big)\Big) \\
&\leq ic(\epsilon)
\end{aligned}
$$

Let $\tilde{c}_1^{(i)}(\epsilon) = c_1(\epsilon) + ic(\epsilon)$ for any arm $k = 1, \ldots, M$, we obtain that

$$\mathbf{E}(N_{i,j}(T)) \leq \tilde{L}_{i,j}(T) + c_1(\epsilon) + ic(\epsilon) = \tilde{L}_{i,j}(T) + \tilde{c}_1^{(i)}(\epsilon) \tag{4}$$

Since regret $R(T)$ satisfies

$$
\begin{aligned}
R(T) &= \sum_{j=2}^{M} \sum_{i=1}^{j-1} \mathbf{E}(\tilde{N}_{i,j}(T))(p^{(i)} - p^{(j)}) \\
&= \sum_{j=2}^{M} \sum_{i=1}^{j-1} \mathbf{E}(N_{i,j}(T))(p^{(i)} - p^{(i+1)}),
\end{aligned}
$$

Substituting equation 4 to the above inequality, we obtain that

$$R(T) \leq (1+\epsilon) \sum_{j=2}^{M} \sum_{i=1}^{j-1} \frac{(p^{(i)} - p^{(i+1)}) \log T}{K(p^{(j)}, p^{(\min(i,|\mathcal{I}|)))}} + \sum_{j=2}^{M} \sum_{i=1}^{j-1} \tilde{c}_1^{(i)}(\epsilon)(p^{(i)} - p^{(i+1)}).$$

Let $C(\epsilon) = \sum_{j=2}^{M} \sum_{i=1}^{j-1} \tilde{c}_1^{(i)}(\epsilon)(p^{(i)} - p^{(i+1)})$, we have $C(\epsilon) = O(\frac{1}{\epsilon^2})$, which completes the proof. ∎

### A.4  Unknown Cool-Down Duration

### A.4.1  Basic Properties

The exploration of cool-down durations for each arm $k$ comprises two phases: the quick jump stage, where the agent employs a bisection algorithm to promptly update the cool-down parameters, and the confirmation stage, where the agent conducts Bernoulli trials to verify the accuracy of $L_u^{(k)}(t)$. For $c \in \{l^{(k)}, \ldots, D\}$, we define a time sequence as a $c$-run of arm $k$ if it represents the collection of time instances $t$ satisfying $L_u^{(k)}(t) = c$. Note that a $c$-run might not be a consecutive sequence. Additionally, throughout the entire process, $c$ might not traverse all values in $l^{(k)}, \ldots, D$ for each arm $k = 1, \ldots, M$. We start with examining the characteristics of the quick jump stage.

**Lemma 4.** *For each arm $k = 1, \ldots, M$, the accumulated number of selects on arm $k$ in the quick jump stage is at most $c_2^{(k)} \triangleq \frac{(D+l^{(k)})(D-l^{(k)}+1)}{2}$.*

*Proof.* For each arm $k = 1, \ldots, M$, suppose the agent is at the quick jump step in $c$-run of arm $k$, where $c \in \{l^{(k)}, \ldots, D\}$. If the agent obtains a 1-reward, then the agent updates the estimated CD upper bound and breaks $c$-run, and thus never enters the confirmation step of $c$-run. And if the agent obtains a 0-reward, then it continues the quick jump step until it has sampled for $L_u^{(k)}(t) - L_{test}^{(k)}(t)$ times. With this in mind, whenever the agent enters a $c$-run for cool-down exploration on arm $k$, it at most samples $c - L_{test}^{(k)}(t)$ times, within

which, the quick jump stages contains at most $c$ time steps as it is a bisection process. Since $c \in \{l^{(k)}, \ldots, D\}$, then the total number of samplings in the quick jump steps in all runs is lower bounded by

$$\sum_{c=l^{(k)}}^{D} c \leq \frac{(D + l^{(k)})(D - l^{(k)} + 1)}{2},$$

which completes the proof. ∎

The following lemma provides an evaluation of the accuracy of the estimated success rate $\tilde{p}^{(k)}(t)$ for each $k = 1, \ldots, M$.

**Lemma 5.** *For any $k = 1, \ldots, M$ and $\epsilon > 0$, there exists a constant $c_3^{(k)}(\epsilon) = O(\frac{1}{\epsilon^2})$, such that*

$$\sum_{t=1}^{T} \mathbf{P}\left(1 - \tilde{p}^{(k)}(t) \geq (1-p)^{\frac{1}{1+\epsilon}}, a(t) = k\right) \leq c_3^{(k)}(\epsilon).$$

*Proof.* Let

$$g_\epsilon = (1-p)^{\frac{1}{1+\epsilon}} - (1-p),$$

it holds that

$$g_\epsilon = (1-p)^{\frac{1}{1+\epsilon}}\left(1 - (1-p)^{\frac{\epsilon}{1+\epsilon}}\right)$$

$$\geq (1-p)^{\frac{1}{1+\epsilon}} p \frac{\epsilon}{1+\epsilon} = O(\epsilon) \tag{5}$$

From Lemma 2, for any arm $k = 1, \ldots, M$, it holds that

$$\mathbf{P}\left(1 - \tilde{p}^{(k)}(t) \geq (1-p)^{\frac{1}{1+\epsilon}}, a(t) = k\right) \leq \exp(-2g_\epsilon^2 |\mathcal{C}_1^{(k)}(t)|).$$

Since for any $t > 0$,

$$\mathbf{P}\left(1 - \tilde{p}^{(k)}(t) \geq (1-p)^{\frac{1}{1+\epsilon}}, a(t) = k\right)$$

$$= \mathbf{P}\left(1 - \tilde{p}^{(k)}(t) \geq (1-p)^{\frac{1}{1+\epsilon}}, t \in \mathcal{C}_1^{(k)}(T)\right) + \mathbf{P}\left(1 - \tilde{p}^{(k)}(t) \geq (1-p)^{\frac{1}{1+\epsilon}}, t \in \mathcal{C}_2^{(k)}(T)\right),$$

then

$$\mathbf{P}\left(1 - \tilde{p}^{(k)}(t) \geq (1-p)^{\frac{1}{1+\epsilon}}, a(t) = k\right)$$

$$\leq \sum_{t \in \mathcal{C}_1^{(k)}(T)} \exp(-2g_\epsilon^2 |\mathcal{C}_1^{(k)}(t)|) + \sum_{t \in \mathcal{C}_2^{(k)}(T)} \exp(-2g_\epsilon^2 |\mathcal{C}_1^{(k)}(t)|). \tag{6}$$

For each $\tau \in \mathcal{C}_2^{(k)}(T)$, let $x_\tau$ be the number of successive steps of the cool-down exploration starting from $\tau$. Specially, we let $x_\tau = 0$ for $\tau \notin \mathcal{C}_2^{(k)}(T)$. Note that for any $\tau \in \mathcal{C}_2^{(k)}(T)$,

$$\max\{t < \tau : a(t) = k\} \in \mathcal{C}_1^{(k)}(T). \tag{7}$$

Then for any $\tau \in \mathcal{C}_2^{(k)}(T)$, $f : \tau \to \arg\max\{t < \tau : a(t) = k\}$ is an injection. Applying this property to equation 6, we obtain that

$$\mathbf{P}\left(1 - \tilde{p}^{(k)}(t) \geq (1-p)^{\frac{1}{1+\epsilon}}, a(t) = k\right)$$

$$\leq \sum_{h=1}^{|\mathcal{C}_1^{(k)}(T)|} \exp(-2g_\epsilon^2 h) + \sum_{\tau \in \mathcal{C}_2^{(k)}(T)} \exp(-2g_\epsilon^2 |\mathcal{C}_1^{(k)}(f(\tau))|) x_\tau$$

$$\leq \sum_{h=1}^{|\mathcal{C}_1^{(k)}(T)|} \exp(-2g_\epsilon^2 h)(1 + \tilde{x}_h),$$

where $\tilde{x}_h$ denotes the number of time instances the agent spends on cool-down exploration between the $h$th and $(h+1)$th distribution explorations. Note that $\tilde{x}_h > 1$ only when the agent enters the quick jump stage for arm $k$ between the $h$th and $(h+1)$th distribution explorations. From Lemma 4, the quick jump stage for arm $k$ takes at most $c_2^{(k)}$ instances, then

$$\sum_{t=1}^{T} \mathbf{P}\left(1 - \tilde{p}^{(k)}(t) \geq (1-p)^{\frac{1}{1+\epsilon}}, a(t) = k\right)$$

$$\leq \exp(-2g_\epsilon^2)(1 + c_2^{(k)}) + 2\sum_{h=1}^{|\mathcal{C}_1^{(k)}(T)|} \exp(-2g_\epsilon^2 h).$$

Then combining the fact that $\sum_{t=1}^{\infty} e^{-ct} = O(\frac{1}{c})$ for any positive constant $c$ and equation 5 to the above inequality, we obtain that

$$\sum_{t=1}^{T} \mathbf{P}\left(1 - \tilde{p}^{(k)}(t) \geq (1-p)^{\frac{1}{1+\epsilon}}, a(t) = k\right) = O(\frac{1}{\epsilon^2}).$$

This implies that there exists a constant $c_3^{(k)}(\epsilon) = O(\frac{1}{\epsilon^2})$, such that

$$\sum_{t=1}^{T} \mathbf{P}\left(1 - \tilde{p}^{(k)}(t) \geq (1-p)^{\frac{1}{1+\epsilon}}, a(t) = k\right) \leq c_3^{(k)}(\epsilon),$$

which completes the proof. ∎

### A.4.2 Insufficient Cool-Down Exploration

As discussed in Section 1.3, sub-linear regret can arise if the optimal active arm is erroneously classified as inactive (not included in the belief arm set $\mathcal{B}(t)$) at each time $t$. In this section, we analyze the likelihood of this occurrence. We investigate this scenario under two distinct conditions: firstly, when $waitlist^{(k)}(t) = 0$, i.e., when the agent dos not pause the cool-down exploration of $k$ at time $t$, and secondly, when $waitlist^{(k)}(t) = 1$, i,e., when the agent has paused the cool-down exploration of $k$ at time $t$.

**Lemma 6.** *For any arm $k = 1, \ldots, M$, it holds that*

$$\sum_{t \in \mathcal{T}_k^o(T)} \mathbf{P}(waitlist^{(k)}(t) = 0, k \notin \mathcal{B}(t)) \leq \frac{(M+1)(D - l^{(k)})}{p^{(k)}}.$$

*Proof.* For any arm $k = 1, \ldots, M$ and time $t$, if the agent enters the cool-down exploration for arm $k$ at time $t$ and $c = L_u^k(t) > L^{(k)} \geq l^{(k)}$, then arm $k$ must be active at time $\tau$, i.e., $t \in \mathcal{T}_k^a(T)$. Thus with probability $p^{(k)}$, the agent breaks $c$-run for arm $k$, and consequently $L_u^k(t+1) < L_u^k(t)$. Since there are at

most $D - l^{(k)}$ runs before $L_u^{(k)}(t)$ reaches $L^{(k)}$, then the expected number of rounds of cool-down exploration is upper bounded by $\frac{D - l^{(k)}}{p^{(k)}}$. Note that after the agent executes a distribution exploration at time $t$, there exists a time $t' \in (t, t + D)$, such that $CD\_Explore^{(k)}(t) = L_{test}^{(k)}(t) - L_u^{(k)}(t) + 1$ whenever $waitlist^{(k)}(t) = 0$. Then from the decision-making process, there must be one cool-down exploration for arm $k$ after at most $M$ distribution explorations for arm $k$. Let $a(t) = 0$ if the agent does not enter the decision-making process at time $t$. Then

$$\mathbf{E}(|\{t : waitlist^{(k)}(t) = 0, L_u^{(k)}(t) > L^{(k)}\}|) \leq \frac{(M+1)(D - l^{(k)})}{p^{(k)}}.$$

From the definition of $L^{(k)}$, it holds for any $k = 1, \ldots, M$ that

$$\{t : k \in \mathcal{A}(t) \cap \mathcal{B}(t)^{\mathbf{c}}\} \subseteq \{t : L_u^{(k)}(t) > L^{(k)}\}. \tag{8}$$

Consequently,

$$\sum_{t \in \mathcal{T}_k^o(T)} \mathbf{P}(waitlist^{(k)}(t) = 0, k \notin \mathcal{B}(t)) \leq \mathbf{E}(|\{t : waitlist^{(k)}(t) = 0, L_u^{(k)}(t) > L^{(k)}\}|)$$

$$\leq \frac{(M+1)(D - l^{(k)})}{p^{(k)}},$$

which completes the proof. ∎

**Lemma 7.** *For any arm $k = 1, \ldots, M$, there exist constants $c_4^{(k)}(\epsilon) = O(\frac{-\log \epsilon}{\epsilon^2}), \tilde{c}_5^{(k)}(\epsilon) = O(1)$, such that*

$$\sum_{t \in \mathcal{T}_k^o(T)} \mathbf{P}(waitlist^{(k)} = 1, k \notin \mathcal{B}(t), t \geq c_4^{(k)}(\epsilon)) \leq \log \log T + c_5^{(k)}(\epsilon).$$

*Proof.* Note that $waitlist^{(k)}(t) = 1$ is equivalent to that the agent has sampled $check^{(k)}(t)$ times in the confirmation step of the $L_u^{(k)}(t)$-run without receiving a 1-reward. Note that if $L_u^{(k)}(t) > l^{(k)}$, then $L_{test}^{(k)}(t) = L_u^{(k)}(t) - 1 \geq l^{(k)}$, implying that arm $k$ is active for any time $t$ on the confirmation stage of an any $c$-run satisfying $c > l^{(k)}$. Consequently

$$\mathbf{P}(L_u^{(k)}(t) > l^{(k)}, waitlist^{(k)} = 1) = (1 - p^{(k)})^{check^{(k)}(t)}, \tag{9}$$

where $check^{(k)}(t)$ satisfies that

$$(1 - \tilde{p}^{(k)}(t))^{check^{(k)}(t)} \leq \frac{1}{t \log t}.$$

We consider the following two cases.

(1) If $p \geq \tilde{p}^{(k)}(t)$, then it holds that

$$(1 - p^{(k)})^{check^{(k)}(t)} \leq (1 - \tilde{p}^{(k)}(t))^{check^{(k)}(t)} \leq \frac{1}{t \log t}. \tag{10}$$

(2) If $p < \tilde{p}^{(k)}(t)$, then

$$(1 - p^{(k)})^{check^{(k)}(t)} \leq (1 - p^{(k)})^{\frac{\log t + \log \log t}{-\log(1 - \tilde{p}^{(k)}(t))}}$$

$$= (1 - \tilde{p}^{(k)}(t))^{\frac{\log t + \log \log t}{-\log(1 - \tilde{p}^{(k)}(t))}} \left(1 + \frac{\tilde{p}^{(k)}(t) - p^{(k)}}{1 - \tilde{p}^{(k)}(t)}\right)^{\frac{\log t + \log \log t}{-\log(1 - \tilde{p}^{(k)}(t))}}.$$

set $a$ in Lemma 2 as $\sqrt{t \log t}$, we have

$$\mathbf{P}\left(\tilde{p}^{(k)}(t) \geq p^{(k)} + \sqrt{\frac{\log t}{t}}\right) \leq \frac{1}{t^2},$$

implying that for any $t$, with probability $1 - \frac{1}{t^2}$,

$$
\begin{aligned}
(1-p)^{check^{(k)}(t)} &\leq \left(1 + \sqrt{\frac{\log t}{t}}/(1 - \tilde{p}^{(k)}(t))\right)^{\frac{\log t + \log\log t}{-\log(1 - \tilde{p}^{(k)}(t))}} (1 - \tilde{p}^{(k)}(t))^{\frac{\log t + \log\log t}{-\log(1 - \tilde{p}^{(k)}(t))}} \\
&\leq \left(1 + \sqrt{\frac{\log t}{t}}/(1 - \tilde{p}^{(k)}(t))\right)^{\frac{\log t + \log\log t}{-\log(1 - \tilde{p}^{(k)}(t))}} /t\log t \\
&\leq \left(1 + \frac{\sqrt{\frac{\log t}{t}}}{1 - p^{(k)} - \sqrt{\frac{\log t}{t}}}\right)^{\frac{\log t + \log\log t}{-\log(1 - p^{(k)})}} /t\log t.
\end{aligned}
$$

Note that $\sqrt{\log t/t}$ converges to 0 when $t$ goes to infinity and that $(1+x)^c \to 1 + cx$ for any $0 < x < 1$ and $c \to \infty$. Then for any $\epsilon$, there exists a constant $c_4^{(k)}(\epsilon) = O(\frac{-\log\epsilon}{\epsilon^2})$, such that when $t \geq c_4^{(k)}(\epsilon)$, there holds

$$
\left(1 + \frac{\sqrt{\frac{\log t}{t}}}{1 - p^{(k)} - \sqrt{\frac{\log t}{t}}}\right)^{\frac{\log t + \log\log t}{-\log(1 - p^{(k)})}} \leq 1 + \frac{1 + \epsilon}{1 - p^{(k)}} \sqrt{\frac{\log t}{t}} \frac{\log t + \log\log t}{-\log(1 - p^{(k)})}.
$$

Combining the above results together, when $p < \tilde{p}^{(k)}(t)$, for any $\epsilon > 0$, given $t \geq c_4^{(k)}(\epsilon)$, with probability $1 - \frac{1}{t^2}$, it holds that

$$
\begin{aligned}
(1-p)^{check^{(k)}(t)} &\leq \frac{1}{t\log t} + \frac{1 + \epsilon}{1 - p^{(k)}} \sqrt{\frac{\log t}{t}} \frac{\log t + \log\log t}{-\log(1 - p^{(k)})} \frac{1}{t\log t} \\
&= \frac{1}{t\log t} + \frac{1 + \epsilon}{-(1 - p^{(k)})\log(1 - p^{(k)})} \frac{\log t + \log\log t}{t\sqrt{t\log t}}.
\end{aligned}
$$

Along with equation 10, we obtain that for any $\epsilon > 0$, given $t \geq c_4^{(k)}(\epsilon)$, with probability at least $1 - \frac{1}{t^2}$, it holds that

$$
(1-p)^{check^{(k)}(t)} \leq \frac{1}{t\log t} + \frac{1 + \epsilon}{-(1 - p^{(k)})\log(1 - p^{(k)})} \frac{\log t + \log\log t}{t\sqrt{t\log t}}.
$$

Then from equation 9, there holds

$$
\sum_{t \in \mathcal{T}_k^o(T)} \mathbf{P}(L_u^{(k)}(t) > l^{(k)}, waitlist^{(k)} = 1, t \geq c_4^{(k)}(\epsilon))
$$

$$
\leq \sum_{t=1}^{T} \left(\frac{1}{t^2} + \frac{1}{t\log t} + \frac{1 + \epsilon}{-(1 - p^{(k)})\log(1 - p^{(k)})} \frac{\log t + \log\log t}{t\sqrt{t\log t}}\right).
$$

Since $\frac{\log t + \log\log t}{t\sqrt{t\log t}} = O(t^{-\frac{3}{2}}\sqrt{\log t})$, it holds that $\sum_{t=1}^{T} \frac{\log t + \log\log t}{t\sqrt{t\log t}} = O(1)$. Then, there exists a constant $c_5^{(k)}(\epsilon) = O(1)$, such that

$$
\sum_{t \in \mathcal{T}_k^o(T)} \mathbf{P}(L_u^{(k)}(t) > l^{(k)}, waitlist^{(k)} = 1, t \geq c_4^{(k)}(\epsilon)) \leq \log\log T + c_5^{(k)}(\epsilon).
$$

Then from equation 8, it holds that

$$
\sum_{t \in \mathcal{T}_k^o(T)} \mathbf{P}(waitlist^{(k)} = 1, k \notin \mathcal{B}(t), t \geq c_4^{(k)}(\epsilon)) \leq \log\log T + c_5^{(k)}(\epsilon),
$$

which completes the proof. ∎

### A.4.3 Sampling Times in Cool-Down Exploration

In this section, we study the expected number of samplings in the cool-down exploration for each arm $k$.

The following lemma provides the upper bound of the number of time instances in the cool-down exploration of arm $k$ for $l^{(k)}$-run.

**Lemma 8.** *For any arm $k = 1, \ldots, M$ and $\epsilon > 0$, it holds that*

$$\mathbf{P}(L_u^{(k)}(t) = l^{(k)}, t \in \mathcal{C}_2^{(k)}(T)) \leq \frac{(1+\epsilon)(\log T + \log \log T)}{-\log(1 - p^{(k)})} + c_3^{(k)}(\epsilon) + l^{(k)},$$

*where $c_3^{(k)}(\epsilon)$ is defined in Lemma 5.*

*Proof.* From the analysis in Lemma 4, the number of time instances in the quick jump step of arm $k$ for $L_u^{(k)}(t) = l^{(k)}$ is $l^{(k)}$. Then for any $k = 1, \ldots, M$, it holds that

$$\mathbf{P}(L_u^{(k)}(t) = l^{(k)}, t \in \mathcal{C}_2^{(k)}(T)) \leq \mathbf{P}(L_u^{(k)}(t) = l^{(k)}, t \in \tilde{\mathcal{C}}_2^{(k)}(T)) + l^{(k)}. \tag{11}$$

Let $\mathcal{E}_1(t)$ denotes the event such that $1 - \tilde{p}^{(k)}(t) \geq (1-p)^{\frac{1}{1+\epsilon}}, a(t) = k$. Expanding the first term of equation 11, we obtain that

$$\mathbf{P}(L_u^{(k)}(t) = l^{(k)}, t \in \tilde{\mathcal{C}}_2^{(k)}(T))$$

$$\leq \mathbf{P}\left(L_u^{(k)}(t) = l^{(k)}, \mathcal{E}_1(t), t \in \tilde{\mathcal{C}}_2^{(k)}(T)\right) + \mathbf{P}\left(L_u^{(k)}(t) = l^{(k)}, \neg\mathcal{E}_1(t), t \in \tilde{\mathcal{C}}_2^{(k)}(T)\right)$$

$$\leq \mathbf{P}(\mathcal{E}_1(t)) + \mathbf{P}(L_u^{(k)}(t) = l^{(k)}, t \in \tilde{\mathcal{C}}_2^{(k)}(T) \,|\, \neg\mathcal{E}_1(t)). \tag{12}$$

From the algorithm,

$$\mathbf{P}(L_u^{(k)}(t) = l^{(k)}, t \in \tilde{\mathcal{C}}_2^{(k)}(T) \,|\, \neg\mathcal{E}_1(t))$$

$$= \mathbf{P}\left(check^{(k)}(t) < \frac{\log t + \log \log t}{-\log(1 - \tilde{p}^{(k)}(t))}, t \in \tilde{\mathcal{C}}_2^{(k)}(T) \,|\, \neg\mathcal{E}_1(t)\right)$$

$$\leq \mathbf{P}\left(check^{(k)}(t) < \frac{(1+\epsilon)(\log t + \log \log t)}{-\log(1 - p^{(k)})}, t \in \tilde{\mathcal{C}}_2^{(k)}(T)\right)$$

$$\leq \mathbf{P}\left(check^{(k)}(t) < \frac{(1+\epsilon)(\log T + \log \log T)}{-\log(1 - p^{(k)})}, t \in \tilde{\mathcal{C}}_2^{(k)}(T)\right).$$

Note that for any adjacent $\tau_1, \tau_2 \in \tilde{\mathcal{C}}_2^{(k)}(T)$ with $\tau_1 < \tau_2$, it holds that $check^{(k)}(\tau_1) + 1 = check^{(k)}(\tau_2)$, which implies that

$$\sum_{t=1}^{T} \mathbf{P}(L_u^{(k)}(t) = l^{(k)}, t \in \tilde{\mathcal{C}}_2^{(k)}(T) \,|\, \neg\mathcal{E}_1(t)))$$

$$\leq \sum_{t=1}^{T} \mathbf{P}\left(check^{(k)}(t) < \frac{(1+\epsilon)(\log T + \log \log T)}{-\log(1 - p^{(k)})}, t \in \tilde{\mathcal{C}}_2^{(k)}(T)\right)$$

$$\leq \frac{(1+\epsilon)(\log T + \log \log T)}{-\log(1 - p^{(k)})}.$$

Applying the above inequality along with Lemma 5 to equation 12, we obtain that

$$\mathbf{P}(L_u^{(k)}(t) = l^{(k)}, t \in \tilde{\mathcal{C}}_2^{(k)}(T)) \leq \frac{(1+\epsilon)(\log T + \log \log T)}{-\log(1 - p^{(k)})} + c_3^{(k)}(\epsilon),$$

substituting which to equation 11, we have

$$\mathbf{P}(L_u^{(k)}(t) = l^{(k)}, t \in \mathcal{C}_2^{(k)}(T)) \leq \frac{(1+\epsilon)(\log T + \log \log T)}{-\log(1 - p^{(k)})} + c_3^{(k)}(\epsilon) + l^{(k)},$$

which completes the proof. ∎

### A.4.4 Extension For Results in Section A.3

Since $\tilde{p}^{(k)}(t)$ is updated only when the agent makes distribution exploration, and the analysis of Theorem 2 is influenced by the accuracy of $\tilde{p}^{(k)}(t)$. we extend the findings outlined in Section A.3 by employing the same analytical approach utilized in the proof of Theorem 2.

**Corollary 1.** *For any $j = 2, \ldots, M$ and $i = 1, \ldots, j-1$, for any $\epsilon > 0$, it holds that*

$$\mathbf{E}(N_{i,j}(T)) \leq \frac{(1+\epsilon)\log T}{K(p^{(j)}, p^{(\min(i,|\mathcal{I}|))})} + \tilde{c}_1^{(k)}(\epsilon),$$

*where $\tilde{c}_1^{(k)}(\epsilon)$ is defined in Theorem 2.*

*Proof.* For any $j = 2, \ldots, M$ and $i = 1, \ldots, j-1$,

$$\mathbf{E}(N_{i,j}(T)) = \sum_{t \in (\cup_{k=1}^i \mathcal{T}_k^o) \cap \mathcal{T}_j^a} \mathbf{P}(t \in \mathcal{C}_1^{(j)}(t)) = \sum_{k=1}^i \sum_{t \in \mathcal{T}_j^a(T) \cap \mathcal{T}_i^o(T)} \mathbf{P}(t \in \mathcal{C}_1^{(j)}(t)). \tag{13}$$

Since $\{t \leq T : a(t) = j\} \supset \mathcal{C}_1^{(j)}(T)$ for any $j = 1, \ldots, M$, there holds

$$\mathbf{E}(N_{i,j}(T)) \leq \sum_{t \in (\cup_{k=1}^i \mathcal{T}_k^o) \cap \mathcal{T}_j^a} \mathbf{P}(a(t) = j) = \sum_{k=1}^i \sum_{t \in \mathcal{T}_j^a(T) \cap \mathcal{T}_i^o(T)} \mathbf{P}(a(t) = j).$$

Then using the exact same technique in the proof of Theorem 2, we obtain that

$$\mathbf{E}(N_{i,j}(T)) \leq \frac{(1+\epsilon)\log T}{K(p^{(j)}, p^{(\min(i,|\mathcal{I}|))})} + \tilde{c}_1^{(i)}(\epsilon),$$

which completes the proof. $\blacksquare$

**Corollary 2.** *For any $i \in \mathcal{I}$, $j \in \{i+1, \ldots, M\}$ and $j > i$, and $\epsilon > 0$, for the same constant defined in Corollary 1, it holds that*

$$\mathbf{E}(M_{i,j}(T)) < \mathbf{E}(N_{i,j}(T)) + c_2^{(j)}.$$

*Proof.* For any $j = 2, \ldots, M$ and $i = 1, \ldots, j-1$, it holds that

$$\begin{aligned}
\mathbf{E}(M_{i,j}(T)) &= \sum_{t \in (\cup_{k=1}^i \mathcal{T}_k^o) \cap \mathcal{T}_j^a} \mathbf{P}(t \in \mathcal{C}_2^{(j)}(T)) x_t^{(j)} \\
&= \sum_{k=1}^i \sum_{t \in \mathcal{T}_j^a(T) \cap \mathcal{T}_i^o(T)} \mathbf{P}(t \in \mathcal{C}_2^{(j)}(T)) x_t^{(j)} \\
&< \sum_{k=1}^i \sum_{t \in \mathcal{T}_j^a(T) \cap \mathcal{T}_i^o(T)} \mathbf{P}(t \in \mathcal{C}_2^{(j)}(T)) + \sum_{t=1}^T x_t^{(j)} \mathbf{1}(x_t^{(j)} > 1),
\end{aligned} \tag{14}$$

where $x_t^{(j)}$ is be the number of steps of the successive cool-down exploration starting from $t$ for any $t > 0$. Specially, $x_t^{(j)} = 0$ for $t \notin \mathcal{C}_2^{(j)}(T)$.

Note that $x_t^{(j)} > 1$ only when $t$ is the starting time of a cool-down exploration on the quick jump stage. Then from Lemma 4, it holds that

$$\sum_{t=1}^T x_t^{(j)} \mathbf{1}(x_t^{(j)} > 1) \leq c_2^{(j)}.$$

Substituting the above inequality to equation 14, we obtain that

$$\mathbf{E}(M_{i,j}(T)) < \sum_{k=1}^{i} \sum_{t \in \mathcal{T}_j^a(T) \cap \mathcal{T}_i^o(T)} \mathbf{P}(t \in \mathcal{C}_2^{(j)}(T)) + c_2^{(j)}.$$

Recall the property introduced in Lemma 5: for any $\tau \in \mathcal{C}_2^{(k)}(T)$, $f : \tau \to \max\{t < \tau : a(t) = k\}$ is an injection, in this sense, since $\tilde{p}^{(j)}(t)$ only changes when $t \in \mathcal{C}_1^{(j)}(t)$ for any $j = 1, \ldots, M$, the above inequality can be further modified as

$$\mathbf{E}(M_{i,j}(T)) < \sum_{k=1}^{i} \sum_{t \in \mathcal{T}_j^a(T) \cap \mathcal{T}_i^o(T)} \mathbf{P}(t \in \mathcal{C}_1^{(j)}(T)) + c_2^{(j)}.$$

Then substituting equation 13 to the above inequality, we obtain that

$$\mathbf{E}(M_{i,j}(T)) < \mathbf{E}(N_{i,j}(T)) + c_2^{(j)},$$

which completes the proof. ∎

With all the above results in hand, we are able to prove Theorem 4.

*Proof of Theorem 4:* For any $k = 1, \ldots, M$ and time $t$ such that $a^*(t) = k$, a loss in reward is likely to incur when the agent fails to select $k$ because

1. $k \in \mathcal{B}(t)$, but there exists an arm $j > k$ such that $\theta^{(j)}(t) > \theta^{(k)}(t)$

2. $k \notin \mathcal{B}(t)$

3. there exists an arm $i < k$, such that $i \in \mathcal{B}(t)$ and the agent selects $i$

4. $k \in \mathcal{B}(t)$, $k = \arg\max_{i \in \mathcal{B}(t)} \theta^{(i)}(t)$, but the agent enters the urgent cool-down exploration phase for some arm $j > k$

We use $R_1(T), R_2(T), R_3(T), R_4(T)$ to denote the total regret incurred in the above four cases respectively and divide the following analysis into three parts.

**Part 1.** For each arm $j = 2, \ldots, M$ and $i = 1, \ldots, j-1$ in case 1 such that $a^*(t) = i$ and $a(t) = j$, it is easy to see that agent $i$ is active. Then if the agent selects arm $j$ for distribution exploration, the one-time instant regret is $p^{(i)} - p^{(j)}$, and if the agent selects arm $j$ for cool-down exploration, the one-time instant regret is at most $p^{(i)}$. Then it holds that

$$R_2(T) \le \sum_{j=2}^{M} \sum_{i=1}^{j-1} \mathbf{E}(\tilde{N}_{i,j}(T))(p^{(i)} - p^{(j)}) + \sum_{j=2}^{M} \sum_{i=1}^{j-1} \mathbf{E}(\tilde{M}_{i,j}(T))p^{(i)}$$

$$= \sum_{j=2}^{M} \sum_{i=1}^{j-1} \mathbf{E}(N_{i,j}(T))(p^{(i)} - p^{(i+1)}) + \sum_{j=2}^{M} \sum_{i=1}^{j-1} \mathbf{E}(\tilde{M}_{i,j}(T))p^{(i)}.$$

Then substituting Corollary 1 to the above inequality, we obtain that

$$R_2(T) \le \sum_{j=2}^{M} \sum_{i=1}^{j-1} \frac{(1+\epsilon)(p^{(i)} - p^{(i+1)}) \log T}{K(p^{(j)}, p^{(\min(i,|\mathcal{I}|))})} + \sum_{j=2}^{M} \sum_{i=1}^{j-1} \tilde{c}_1^{(i)}(\epsilon) + \sum_{j=2}^{M} \sum_{i=1}^{j-1} \mathbf{E}(\tilde{M}_{i,j}(T))p^{(i)}.$$

**Part 2.** In case 2, for each arm $k$, and $a^*(t) = k$, the one-time regret is at most $p^{(k)}$, then

$$R_1(T) \le \sum_{k=1}^{M} p^{(k)} \sum_{t \in \mathcal{T}_k^o(T)} \mathbf{P}(k \notin \mathcal{B}(t))$$

$$= \sum_{k=1}^{M} p^{(k)} \left( \sum_{t \in \mathcal{T}_k^o(T)} \mathbf{P}(waitlist^{(k)}(t) = 0, k \notin \mathcal{B}(t)) + \sum_{t \in \mathcal{T}_k^o(T)} \mathbf{P}(waitlist^{(k)} = 1, k \notin \mathcal{B}(t)) \right).$$

From Lemmas 6 and 7, we obtain that

$$R_1(T) \leq \sum_{k=1}^{M} p^{(k)} \left( c_4^{(k)}(\epsilon) + \sum_{t \in \mathcal{T}_k^o(T)} \mathbf{P}(waitlist^{(k)} = 1, k \notin \mathcal{B}(t), t \geq c_4^{(k)}(\epsilon)) \right)$$

$$+ \sum_{k=1}^{M} p^{(k)} \frac{(M+1)(D - l^{(k)})}{p^{(k)}}$$

$$\leq M \log \log T + \sum_{k=1}^{M} (M+1) \left( (D - l^{(k)}) + p^{(k)}(c_4^{(k)}(\epsilon) + c_5^{(k)}(\epsilon)) \right).$$

**Part 3.** For any $j = 2, \ldots, M$ and $i = 1, \ldots, j-1$, let $n_3(i,j)$ be the number of selects on arm $i$ when $j$ is the actual optimal active arm, as described in case 3, and $n_3(i) = \sum_{j=1}^{i-1} n_3(i,j)$. In such a situation, since $j$ is active, the one time instant regret is at most $p^{(j)}$. In this sense, it holds that

$$R_3(T) \leq \sum_{j=2}^{M} \sum_{i=1}^{j-1} \mathbf{E}(n_3(i,j)) p^{(j)}$$

$$\leq \sum_{j=2}^{M} \sum_{i=1}^{j-1} \mathbf{E}(n_3(i,j)) p^{(i)}$$

$$= \sum_{j=2}^{M} \mathbf{E}(n_3(j)) p^{(j)}.$$

**Part 4.** For any $j = 2, \ldots, M$ and $i = 1, \ldots, j-1$, let $n_4(i,j)$ denote the number of selects on arm $j$ when arm $i$ is the optimal arm in case 4. Then

$$R_4(T) \leq \sum_{j=2}^{M} \sum_{i=1}^{j-1} \mathbf{E}(n_4(i,j)) p^{(i)}.$$

Note that according to the urgent decision-making requirement, case 4 only happens when the latest cool-down exploration on arm $j$ before time $t$ is earlier than the that on arm $i$. Then it holds for any $j = 2, \ldots, M$ and $i = 1, \ldots, j-1$ that

$$n_4(i,j) \leq m^{(i)}(t). \tag{15}$$

Adding all the "partial" regret together, we obtain that

$$R(T) \leq M \log \log T + \sum_{k=1}^{M} (M+1) \left( (D - l^{(k)}) + p^{(k)}(c_4^{(k)}(\epsilon) + c_5^{(k)}(\epsilon)) \right)$$

$$+ \sum_{j=2}^{M} \sum_{i=1}^{j-1} \frac{(1+\epsilon)(p^{(i)} - p^{(i+1)}) \log T}{K(p^{(j)}, p^{(\min(i, |\mathcal{I}|))})} + \sum_{j=2}^{M} \sum_{i=1}^{j-1} \tilde{c}_1^{(i)}(\epsilon) + \sum_{j=2}^{M} \sum_{i=1}^{j-1} \mathbf{E}(\tilde{M}_{i,j}(T)) p^{(i)}$$

$$+ \sum_{j=2}^{M} \mathbf{E}(n_3(j)) p^{(j)} + \sum_{j=2}^{M} \sum_{i=1}^{j-1} \mathbf{E}(n_4(i,j)) p^{(i)}. \tag{16}$$

Note that for any $j = 2, \ldots, M$, it holds that

$$m^{(j)}(T) \geq \sum_{i=1}^{j-1} \tilde{M}_{i,j}(T) + n_3(j) + \sum_{i=1}^{j-1} n_4(i,j). \tag{17}$$

Then

$$\sum_{i=1}^{j-1} \tilde{M}_{i,j}(T)p^{(i)} + n_3(j)p^{(j)} + \sum_{i=1}^{j-1} n_4(i,j)p^{(i)}$$

$$\leq \sum_{i=1}^{j-1} \tilde{M}_{i,j}(T)p^{(i)} + \sum_{i=1}^{j-1} n_4(i,j)p^{(i)} + \left( m^{(j)}(T) - \sum_{i=1}^{j-1} (\tilde{M}_{i,j}(T) + n_4(i,j)) \right) p^{(j)}$$

$$= \sum_{i=1}^{j-1} (\tilde{M}_{i,j}(T) + n_4(i,j))(p^{(i)} - p^{(j)}) + m^{(j)}(T)p^{(j)}.$$

Substituting equation 15 to the above inequality, we obtain that

$$\sum_{i=1}^{j-1} \tilde{M}_{i,j}(T)p^{(i)} + n_3(j)p^{(j)} + \sum_{i=1}^{j-1} n_4(i,j)p^{(i)}$$

$$\leq \sum_{i=1}^{j-1} \tilde{M}_{i,j}(T)(p^{(i)} - p^{(j)}) + \sum_{i=1}^{j-1} m^{(i)}(T)(p^{(i)} - p^{(j)}) + m^{(j)}(T)p^{(j)}$$

$$\leq \sum_{i=1}^{j-1} \tilde{M}_{i,j}(T)(p^{(i)} - p^{(j)}) + \sum_{i=1}^{j} m^{(i)}(T)p^{(i)}.$$

Furthermore, equation 17 also indicates that

$$\sum_{i=1}^{j-1} \tilde{M}_{i,j}(T)p^{(i)} + n_3(j)p^{(j)} + \sum_{i=1}^{j-1} n_4(i,j)p^{(i)}$$

$$\leq \sum_{i=1}^{j-1} \tilde{M}_{i,j}(T)p^{(i)} + n_3(j)p^{(i)} + \sum_{i=1}^{j-1} n_4(i,j)p^{(i)}$$

$$\leq m^{(j)}(T)p^{(i)}.$$

Then together it holds for all $j = 2, \ldots, M$ that

$$\sum_{i=1}^{j-1} \tilde{M}_{i,j}(T)p^{(i)} + n_3(j)p^{(j)} + \sum_{i=1}^{j-1} n_4(i,j)p^{(i)}$$

$$\leq \min \left\{ m^{(j)}(T)p^{(i)}, \sum_{i=1}^{j-1} \tilde{M}_{i,j}(T)(p^{(i)} - p^{(j)}) + \sum_{i=1}^{j} m^{(i)}(T)p^{(i)} \right\}.$$

Summing the above inequality up for $j = 2, \ldots, M$, we obtain that

$$\sum_{j=2}^{M} \sum_{i=1}^{j-1} \mathbf{E}(\tilde{M}_{i,j}(T))p^{(i)} + \sum_{j=2}^{M} \mathbf{E}(n_3(j))p^{(j)} + \sum_{j=2}^{M} \sum_{i=1}^{j-1} \mathbf{E}(n_4(i,j))p^{(i)}$$

$$\leq \min \left\{ \sum_{j=2}^{M} \mathbf{E}(m^{(j)}(T))p^{(i)}, \sum_{j=2}^{M} \sum_{i=1}^{j-1} \mathbf{E}(\tilde{M}_{i,j}(T))(p^{(i)} - p^{(j)}) + \sum_{j=2}^{M} \sum_{i=1}^{j} \mathbf{E}(m^{(i)}(T))p^{(i)} \right\}$$

$$= \min \left\{ \sum_{j=2}^{M} \mathbf{E}(m^{(j)}(T))p^{(i)}, \sum_{j=2}^{M} \sum_{i=1}^{j-1} \mathbf{E}(M_{i,j}(T))(p^{(i)} - p^{(i+1)}) + \sum_{j=2}^{M} \sum_{i=1}^{j} \mathbf{E}(m^{(i)}(T))p^{(i)} \right\}$$

$$\leq \min \left\{ \sum_{j=2}^{M} \mathbf{E}(m^{(j)}(T))p^{(i)}, \sum_{j=2}^{M} \sum_{i=1}^{j-1} \mathbf{E}(N_{i,j}(T))(p^{(i)} - p^{(i+1)}) + \sum_{j=2}^{M} \sum_{i=1}^{j} \mathbf{E}(m^{(i)}(T))p^{(i)} + \sum_{j=2}^{M} \sum_{i=1}^{j-1} c_2^{(j)} \right\},$$

where the last step is from Corollary 2. From Lemma 8 and the analysis of Lemma 6, it holds for all $j = 1, \ldots, M$ that

$$\mathbf{E}(m^{(j)}(T)) \leq \frac{(1+\epsilon)(\log T + \log \log T)}{-\log(1 - p^{(j)})} + c_3^{(j)}(\epsilon) + l^{(j)} + \frac{D - l^{(j)}}{p^{(j)}}.$$

Then

$$\sum_{j=2}^{M} \sum_{i=1}^{j-1} \mathbf{E}(\tilde{M}_{i,j}(T))p^{(i)} + \sum_{j=2}^{M} \mathbf{E}(n_3(j))p^{(j)} + \sum_{j=2}^{M} \sum_{i=1}^{j-1} \mathbf{E}(n_4(i,j))p^{(i)}$$

$$\leq \min \left\{ \sum_{j=2}^{M} \sum_{i=1}^{j-1} \mathbf{E}(N_{i,j}(T))(p^{(i)} - p^{(i+1)}) + \sum_{j=2}^{M} \sum_{i=1}^{j} \frac{(1+\epsilon)p^{(i)}}{-\log(1 - p^{(i)})}(\log T + \log \log T), \right.$$

$$\left. \sum_{j=2}^{M} \frac{(1+\epsilon)p^{(i)}}{-\log(1 - p^{(j)})}(\log T + \log \log T) \right\} + \sum_{j=2}^{M} \sum_{i=1}^{j} \left( c_3^{(j)}(\epsilon) + l^{(j)} + \frac{D - l^{(j)}}{p^{(j)}} + c_2^{(j)} \right)$$

$$\leq \min \left\{ \sum_{j=2}^{M} \sum_{i=1}^{j-1} \frac{(1+\epsilon)(p^{(i)} - p^{(i+1)}) \log T}{K(p^{(j)}, p^{(\min(i,|\mathcal{I}|))})} + \sum_{j=2}^{M} \sum_{i=1}^{j} \frac{(1+\epsilon)p^{(i)}}{-\log(1 - p^{(i)})}(\log T + \log \log T), \right.$$

$$\left. \sum_{j=2}^{M} \frac{(1+\epsilon)p^{(i)}}{-\log(1 - p^{(j)})}(\log T + \log \log T) \right\} + \sum_{j=2}^{M} \sum_{i=1}^{j} \left( c_3^{(j)}(\epsilon) + l^{(j)} + \frac{D - l^{(j)}}{p^{(j)}} + c_2^{(j)} \right).$$

Substituting the above inequality to equation 16, we obtain that

$$R(T) \leq \min \left\{ (1+\epsilon) \sum_{j=2}^{M} \sum_{i=1}^{j-1} \frac{2(p^{(i)} - p^{(i+1)}) \log T}{K(p^{(j)}, p^{(\min(i,|\mathcal{I}|))})} + \sum_{j=2}^{M} \sum_{i=1}^{j} \frac{(1+\epsilon)p^{(i)}}{-\log(1 - p^{(i)})} \log T, \right.$$

$$\left. (1+\epsilon) \sum_{j=2}^{M} \sum_{i=1}^{j-1} \frac{(p^{(i)} - p^{(i+1)}) \log T}{K(p^{(j)}, p^{(\min(i,|\mathcal{I}|))})} + \sum_{j=2}^{M} \frac{(1+\epsilon)p^{(i)}}{-\log(1 - p^{(j)})} \log T \right\}$$

$$+ \hat{C}_1(\epsilon) \log \log T + \hat{C}_2(\epsilon),$$

where

$$\hat{C}_1(\epsilon) = M + \min \left\{ \sum_{j=2}^{M} \sum_{i=1}^{j} \frac{(1+\epsilon)p^{(i)}}{-\log(1 - p^{(i)})}, \sum_{j=2}^{M} \frac{(1+\epsilon)p^{(i)}}{-\log(1 - p^{(j)})} \right\} = O(1),$$

and $\hat{C}_2(\epsilon)$ is the sum of all the constant terms. It can be obtained from Lemmas 4-8 that $\hat{C}_2(\epsilon) = O(\frac{-\log \epsilon}{\epsilon^2})$. Note that $\frac{x}{-\log(1-x)} \leq 1$ for any $x \in [0, 1]$, then $R(T)$ can be further bounded as

$$R(T) \leq (1+\epsilon) \min \left\{ \sum_{j=2}^{M} \left( \sum_{i=1}^{j-1} \left( \frac{2(p^{(i)} - p^{(i+1)})}{K(p^{(j)}, p^{(\min(i,|\mathcal{I}|))})} + 1 \right) + 1 \right), \right.$$

$$\left. \sum_{j=2}^{M} \sum_{i=1}^{j-1} \frac{(p^{(i)} - p^{(i+1)})}{K(p^{(j)}, p^{(\min(i,|\mathcal{I}|))})} + \sum_{j=2}^{M} \frac{(1+\epsilon)p^{(i)}}{-\log(1 - p^{(j)})} \right\} \log T$$

$$+ \hat{C}_1(\epsilon) \log \log T + \hat{C}_2(\epsilon),$$

which completes the proof. ∎

## B  Supplementary Pseudocode

---

**Algorithm 3:** Distribution Exploration

**Input:** time $t$, $\alpha^{(k)}, \beta^{(k)}, CD\_Explore^{(k)}, L_{test}^{(k)}$ for all $k = 1, \ldots, M$, selected arm $a(t)$

**1** $CD\_Explore^{(k)} = CD\_Explore^{(k)} - 1$ for $k = 1, \ldots, M$

**2** **if** $X_{a(t)} = 1$ **then**

**3** $\quad$ $\alpha^{(a(t))} = \alpha^{(a(t))} + 1$

**4** $\quad$ $CD\_Explore^{(a(t))} = L_{test}^{(a(t))}$ $\qquad\qquad$ // reset cool-down state after obtaining a 1-reward

**5** **else**

**6** $\quad$ $\beta^{(a(t))} = \beta^{(a(t))} + 1$

**7** **end**

---

**Algorithm 4:** Cool-Down Exploration

**Input:** time $t$, arm $a(t)$, $CD\_Explore^{(k)}, L_u^{(k)}, L_{test}^{(k)}, \tilde{L}_{test}^{(k)}, check^{(k)}$ for all $k = 1, \ldots, M$

**1** $t = t - 1$

**2** $\text{upper} = L_u^{(a(t))} - L_{test}^{(a(t))} + CD\_Explore^{(a(t))}$ $\quad$ // largest remaining time before the arm is removed from $\mathcal{B}$

**3** $r1 = 0$ $\qquad\qquad\qquad\qquad$ // defaulted as "receive no 1-reward in the cool-down exploration"

**4** **for** $i = 1 : \text{upper}$ **do**

**5** $\quad$ $t = t + 1$

**6** $\quad$ **if** $t > T$ **then**

**7** $\quad\quad$ **break**

**8** $\quad$ **end**

**9** $\quad$ $CD\_Explore^{(k)} = CD\_Explore^{(k)} - 1$ for all $k = 1, \ldots, M$

**10** $\quad$ **if** $X_{a(t)} = 1$ **then**

**11** $\quad\quad$ $r1 = 1$ $\qquad\qquad\qquad\qquad$ // already receive 1-reward in the cool-down exploration

**12** $\quad\quad$ $L_u^{(a(t))} = L_u^{(a(t))} - \text{upper} + i - 1$ $\qquad\qquad\qquad$ // update of the estimate

**13** $\quad\quad$ $check^{(a(t))} = 0$ $\qquad\qquad\qquad\qquad\qquad$ // reset count

**14** $\quad\quad$ **if** $L_u^{(a(t))} = L_{test}^{(a(t))}$ **then**

**15** $\quad\quad\quad$ $L_{test}^{(a(t))} = \max(0, \min(L_u^{(a(t))} - 1, \tilde{L}_{test}^{(a(t))}))$

**16** $\quad\quad\quad$ $\tilde{L}_{test}^{(a(t))} = 0$

**17** $\quad\quad$ **else**

**18** $\quad\quad\quad$ $L_{test}^{(a(t))} = \min\left( \lceil \frac{L_{test}^{(a(t))} + \tilde{L}_{test}^{(a(t))}}{2} \rceil, L_u^{(a(t))} - 1 \right)$

**19** $\quad\quad$ **end**

**20** $\quad\quad$ $CD\_Explore^{(a(t))} = L_{test}^{(a(t))}$ $\qquad\qquad\qquad\qquad$ // reset cool-down state

**21** $\quad\quad$ **break**

**22** $\quad$ **end**

**23** **end**

**24** **if** $r1 = 0$ $\qquad\qquad\qquad\qquad$ // fail to receive 1 reward in the current cool-down exploration

**25** **then**

**26** $\quad$ $\tilde{L}_{test}^{(a(t))} = L_{test}^{(a(t))}$

**27** $\quad$ $L_{test}^{(a(t))} = \max\left( \lceil \frac{L_{test}^{(a(t))} + L_u^{(a(t))} - \text{upper} + i - 1}{2} \rceil, \min(L_u^{(a(t))} - 1, L_{test}^{(a(t))} + 1) \right)$

**28** $\quad$ $check^{(a(t))} = check^{(a(t))} + 1$ $\qquad\qquad\qquad\qquad$ // update count

**29** **end**

---

**Algorithm 5:** Random Sample

---

**Input:** time $t$, $\alpha^{(k)}, \beta^{(k)}, CD\_Explore^{(k)}, L_u^{(k)}, L_{test}^{(k)}, \tilde{L}_{test}^{(k)}, waitlist^{(k)}$ for all $k = 1, \ldots, M$

   /* If belief arm set is empty, the agent picks a random arm                             */

**1** **if** $\{k \in \{1, \ldots, M\} | \, waitlist^{(k)} = 0\} \neq \emptyset$ **then**

**2**     the agent randomly picks an arm $a(t)$ from $\{k \in \{1, \ldots, M\} | \, waitlist^{(k)} = 0\}$

**3** **else**

**4**     the agent randomly picks an arm $a(t)$ from $\{1, \ldots, M\}$

**5** **end**

**6** $CD\_Explore^{(k)} = CD\_Explore^{(k)} - 1$ for all $k = 1, \ldots, M$

**7** **if** $X_{a(t)} = 1$ **then**

**8**     $L_u^{(a(t))} = L_{test}^{(a(t))} - CD\_Explore^{(a(t))}$

**9**     $L_{test}^{(a(t))} = \max(\lceil \frac{L_u^{(a(t))}}{2} \rceil, \min(\tilde{L}_{test}^{(a(t))}, L_u^{(a(t))} - 1))$

**10**     $CD\_Explore^{(a(t))} = L_{test}^{(a(t))}$

**11** **end**

---

**Algorithm 6:** Post-Update

---

**Input:** time $t$, $\alpha^{(k)}, \beta^{(k)}, CD\_Explore^{(k)}, L_u^{(k)}, L_{test}^{(k)}, \tilde{L}_{test}^{(k)}, waitlist^{(k)}$ for all $k = 1, \ldots, M$

   /* Update of *waitlist*                                                             */

**1** **for** $k = 1, \ldots, M$ **do**

**2**     $\tilde{p}^{(k)} = \frac{\alpha^{(k)}}{\alpha^{(k)} + \beta^{(k)}}$

**3**     **if** $L_u^{(k)} = 0$                             // when cool-down duration is 0

**4**     **then**

**5**         $L_{test}^{(k)} = CD\_Explore^{(k)} = 0$

**6**         $check^{(k)} = \infty$

**7**         $waitlist^{(k)} = 1$                  // terminate cool-down exploration

**8**     **end**

**9**     **if** $waitlist^{(k)} = 1$ *and* $check^{(k)} \neq \infty$ *and* $(1 - \tilde{p}^{(k)})^{check^{(k)}} > \frac{1}{t \log t}$  // when cool-down exploration turns insufficient

**10**     **then**

**11**         $waitlist^{(k)} = 0$                  // restart cool-down exploration

**12**         $L_{test}^{(k)} = L_u^{(k)} - 1$

**13**         $CD\_Explore^{(k)} = CD\_Explore^{(k)} - 1$

**14**     **end**

**15**     **if** $waitlist^{(k)} = 0$ *and* $(1 - \tilde{p}^{(k)})^{check^{(k)}} \leq \frac{1}{t \log t}$      // when cool-down exploration is sufficient

**16**     **then**

**17**         $waitlist^{(k)} = 1$                  // pause cool-down exploration

**18**         $CD\_Explore^{(k)} = CD\_Explore^{(k)} + L_u^{(k)} - L_{test}^{(k)}$

**19**         $L_{test}^{(k)} = L_u^{(k)}$

**20**     **end**

**21** **end**

   /* Update of *CD_Explore*                                                  */

**22** **for** $k = 1, \ldots, M$ **do**

**23**     **if** $CD\_Explore^{(k)} + L_u^{(k)} - L_{test}^{(k)} \leq 0$           // when $k$ is bound to be active

**24**     **then**

**25**         $CD\_Explore^{(k)} = \infty$                  // reset cool-down state

**26**     **end**

**27** **end**