# OpenReview forum: "Thompson Sampling For Bandits With Cool-Down Periods"
_TMLR — Accepted by TMLR_

### Review · Reviewer_Yg2p · 2025-06-12

**Summary Of Contributions:**

The paper works on an interesting problem of bandits with cool down periods where an arms become inactive for a possibly unknown bounded duration after playing and receiving a reward. The problem is useful with various applications as noted by the authors in cognitive radio networks, etc. to online advertising where showing a recommendation may not result till certain time in future as the user already interacted and now will not interact till a certain unknown cool down period. This setup is different from varying reward distribution literature as the later typically works with a fixed number of distribution changes or the distribution changes are continuous. The authors also compare with with sleeping bandits, a similar framework, where a known subset of arms is available for the agent.

Now, the agent, along with learning the underlying reward distribution, also has to learn the cool down period. The same bandit explore-exploit dilemma applies for learning cool down period as well. For this setup, the authors provide a Thompson Sampling based algorithm with carefully constructed cool down exploration to learn the cool down period for the arms. The algorithm conducts urgent cool down exploration when the last cool down explore is not conducted from a sufficient time. Also the algorithm uses bisection search to divide the exploration of cool down period in available domain.

Further, the authors conduct regret analysis of the algorithm and prove that the algorithm has logarithmic regret growth.

**Audience:**

Yes

**Claims And Evidence:**

Yes

**Requested Changes:**

Address the two weaknesses.

**Strengths And Weaknesses:**

Strengths:
1. Novel setup which is well motivated and useful in various applications.
2. Novel analysis which improves on existing literature.

Weaknesses:
1. The algorithm is extremely hard to parse. Even after multiple readings, I found it difficult to follow. I would urge authors to added a detailed description of the intended effects of steps of Algorithm 4/5/6 in the Appendix. For example, authors wrote in Section 3.2 "The quick jump stage occurs when $L^{(k)}_{test}(t) < L^{(k)}_u(t) - 1". What does this check do? How does it help the algorithm.

2. Comparison with Sleeping bandits algorithm of Chatterjee et al. How is the Algorithm 1 different from that provided by Chatterjee et al.? The two algorithms both use Thompson sampling, with selecting arms only from the current active set, which is known to both algorithms because of known cool down periods.

---

> ### Author Response · Authors · 2025-08-26
> **Reply to Reviewer Yg2p**
>
> **More Detailed Explanation:**
> Thank you for your positive feedback and valuable suggestions. We provide below a more detailed explanation of our ideas and algorithms.
>
> 1. Algorithm 4 is about the cool-down exploration process. The cool-down exploration occurs when the agent believes the chosen arm is in its cool-down period. $L_u^{(k)}(t)$ is the conservative estimate of the cool-down duration which is ensured to be no larger than the actual one. And to test its accuracy, we need to select the arm before the end of the belief cool-down period. If we receive a 1-reward, it means the arm is already active, implying the $L_u^{(k)}(t)$ is overestimated, and thus should update it accordingly (lines 10-12 in Algorithm 4). And if we fail to receive a 1-reward, since we do not know whether the 0-reward is an actual reward or a penalty due to arm being inactive, we do not update $L_u^{(k)}(t)$, but add the confidence count ${check}^{(a(t))}$ (line 28 in Algorithm 4). The larger the variable is, the more confident the agent believes $L_u^{(k)}(t)$ is the actual duration (since the confidence score is linked to the belief in $L_u^{(k)}(t)$, it is reset whenever $L_u^{(k)}(t)$ is updated, see line 13 in Algorithm 4). Regarding to the timing and length that we do the above test in cool-down exploration, we introduce an intermediate variable $L_{test}^{(a(t))},$ which is updated gradually approaching $L_u^{(k)}(t)$ to narrow the exploration process:  In Algorithm 4, we use lines 14-18 to efficiently update it: if receive a 1-reward, update according to $L_u^{(k)}(t)$, if no 1-reward is received, update using a bisection algorithm.
>
> 2. We mentioned confidence count ${check}^{(a(t))}$ in the preceding paragraph, which is accumulated when we failed to receive 1-reward during cool-down exploration. We keep monitoring this value in Algorithm 6 to decide whether the agent is confident enough for $L_u^{(k)}(t)$ at time $t$. When the agent is sufficiently confident (line 15), we temporarily suspend the cool-down exploration of that arm, and when it turns insufficient again (line 9), we restart the cool-down exploration. By doing so, we ensure a very low probability of insufficient cool-down exploration, see the analysis in Appendix A4.2, while preventing excessive exploration, as detailed in Lemma 8. The other updates, such as ${waitlist}^{(k)}(t)$: is a signal indicating whether the cool-down exploration of arm $k$ is suspended or not, *CD_Explore*^(k): a variable used for determining whether an arm is in the belief arm set and what kind of exploration (distribution/cool-down exploration) should be taken.
>
> 3. As for what you mentioned, $L^{(k)}_{test}(t) < L^{(k)}_u(t) - 1$ is actually not a requirement, but a stage of the cool-down exploration.
>
> The updating rule of $L_{test}^{(a(t))}$ ensures there are only two cases in the cool-down exploration phase:
>
> (a). $L_{test}^{(a(t))} < L_u^{(k)}(t) - 1,$
> (b). $L_{test}^{(a(t))} = L_u^{(k)}(t) - 1.$
>
> We call the corresponding stages as *quick jump stage* and *confirmation stage* respectively in the analysis, for a distinction of different properties during the two stages.
>
> In the quick jump stage, the bisection updating policy ensures a quick update for $L_{test}^{(a(t))}$, which saves the exploration rounds.
>
> In the confirmation stage, we generally accumulate the confidence count ${check}^{(a(t))}$, which as the second paragraph explains, is used to suspend or restart the cool-down exploration of an arm.
>
> **Regarding the comparison**
> In sleeping bandit scenario, our provided algorithm is actually equivalent to the existing Thompson Sampling one. However, we provided a more advanced analysis that makes fully use of the architecture of the Thompson sampling algorithm (Lemma 3), which ends out a much better theoretical bound compared with the existing work: the dominant $\log T$ term is almost $1/64$ of the existing algorithm! See detailed comparison in Remark 2 in Section 4.2.

---

### Review · Reviewer_ywM1 · 2025-06-16

**Summary Of Contributions:**

The authors study a dynamical bandit environment that is characterized by the periodical availability of the arms. In particular, they focus on crafting  Thompson Sampling-like algorithms that are capable to handle such an environment, ensuring low regret. They also prove the the performance surpasses that of state-of-the-art algorithms in the sleeping bandit scenario. Finally, they provide numerical simulations to validate the theoretical findings.

**Audience:**

Yes

**Broader Impact Concerns:**

I do not think the paper needs a broader impact statement.

**Claims And Evidence:**

Yes

**Requested Changes:**

The concerns about Lemma 3 must be addressed clearly and rigorously, the minor spelling mistakes should be revised as also the presentation of the proofs.

**Strengths And Weaknesses:**

The paper is well motivated, and the scope of the research is clearly defined. Thompson Sampling is a particularly interesting algorithm to study due to the inherent complexity of its theoretical analysis, so any meaningful contribution in this area is valuable to the literature. I appreciated the effort made to list real-world scenarios where the proposed setting can model the environment effectively, such as radio networks, coupon websites, and gaming help ground the theory in practical relevance. The structure of the paper is clear and coherent. I liked how the authors summarized the main technical challenges within the main text, making clearer the contributions and novelty wrt to the existing body of knowldege and the possible impact to the literature also for what regards the theoretical part. The algorithms are well explained, and the notation is clear throughout. The results presented are strong and noteworthy, especially since they match the $O(log(T))$ regret bound. I also have particularly appreciated the discussion and analysis of the unknown cool-down duration setting, which in my view is essential for the real-world applicability of the model, and also conceptually as it explain how the algorithm behave when some parameters of particular importance are not known. However, I have serious concerns about the soundness of some of the results. In particular, I question the validity of Lemma 3 in the preliminaries. The authors claim that this lemma can be inferred from the works of Agrawal and Goyal, but from my understanding, this seems to be a potential misinterpretation. Specifically, it appears the authors conflate the concept of a round $t$, which is purely temporal, with the number of times an arm is selected. They seem to suggest that, on average, the first $L_i(T)$ selections of arm $i$ occur within the first $L_i(T)$ rounds. This seems implausible. Since $L_i(T)$ is not a random variable, the only scenario where this would be consistently true is one in which the algorithm selects in every possible realization the $i$-th arm $L_i(T)$ times in the first $L_i(T)$ rounds, that is impossible. This flawed reasoning  underlies also to the proof of Theorem 2, and therefore impacts the validity of the subsequent corollaries derived from it. Given its foundational role, I believe it is crucial for the authors to clarify precisely how Lemma 3 can be justified, or revise the arguments if a mistake is present. Additionally, I noticed some minor spelling mistakes in the paper’s figures (e.g., “reget” instead of “regret”), which should be corrected. Lastly, while not critical, I would suggest improving the presentation of the proofs section, especially given the paper’s heavy notation. For instance, in the section of Lemma 7, the derivations are quite dense and could benefit from clearer exposition and more step-by-step explanations.

# After the discussion:
In the discussion the authors were able to solve the issues with the proofs, providing corrected explanations for their lemmas. I now think there is enough mathematical evidence to support the claims.

---

> ### Author Response · Authors · 2025-08-26
> **Reply to Reviewer ywM1**
>
> Thank you for your positive feedback and careful check! It turns out to be an issue in writing. Like you pointed out, the conclusion of Lemma 3 should be $\sum_{t=1}^T\mathbf{P}(a(t)=k, n_k(t)\ge L_k(T)+1)\le c(\epsilon),$ where $n_k(t)$ is the number of times selecting a sub-optimal arm $k.$ The lemma can be roughly interpreted as: after selecting a sub-optimal arm for a number of times, the probability of selecting it is negligibly small. Theorem 2 uses the similar logic, the inequality after we referenced Lemma 3 in the proof should be modified as
> \begin{align*}
>     \sum_{t\in\mathcal T_j^a(T)\cap\mathcal T_i^o(T)}\mathbf{P}(a(t)=j, n_j(t)> L_{i,j}(T))\le c(\epsilon).
> \end{align*}
> Then we rewrite eq.(4) as
> \begin{align*}
>     \mathbf{E}(N_{i,j}(T))&\le \mathbf{E}\Big(N_{i,j}(T),n_j(T)\le \max_{k=1,\ldots,i}L_{i,j}(T)\Big)+\mathbf{E}\Big(N_{i,j}(T),n_j(T)>\max_{k=1,\ldots,i}L_{i,j}(T)\Big).
> \end{align*}
> Since $N_{i,j}(T)\le n_j(T)$ from definition, the former term $\le\max_{k=1,\ldots,i}L_{i,j}(T)\le \max(\tilde L_{i,j}(T), c_1(\epsilon))$. Using the similar equality right before eq.(4), the latter term is equivalent to $$\sum_{k=1}^i\sum_{t\in\mathcal T_j^a(T)\cap\mathcal T_i^o(T)} \mathbf{P}\Big(a(t)=j,n_j(T)>\max_{k=1,\ldots,i}L_{i,j}(T)\Big),$$ which $\le \sum_{k=1}^i\sum_{t\in\mathcal T_j^a(T)\cap\mathcal T_i^o(T)} \mathbf{P}\Big(a(t)=j,n_j(T)> L_{i,j}(T)\Big)\le ic(\epsilon)$. The above analysis ensures $\mathbf{E}\Big(N_{i,j}(T))\le \tilde L_{i,j}(T)+c_1(\epsilon)+ic(\epsilon)\Big)$, which is exactly the conclusion in eq.(4). Then by using the same analysis after eq. (4), we can complete the proof of Theorem 2.
>
> We will update the statement and proof in the final submission.

---

> > ### Comment · Reviewer_ywM1 · 2025-08-26
> >
> > I have appreciated your constructive reply, the statement you are presenting now seems more plausible, however I have still some doubts that can be trivially retrieved from Lemma 2.10, 2.11 and 2.12 of Agrawal, more specifically Lemma 2.12 bounds the term $\sum_{t \in T} \mathbf{P}(a(t)=j, n_j(t)> L_{i}(T), \overline{E_i^{\theta}}, E_i^{\mu})$ i.e. is conditioned to other events happening, while in Lemma 2.10 is not conditioned on the event the the suboptimal arm $k$ has been played more than a certain amount of pulls, but on the event that the optimal arm has been played a certain amount of time, so that formally when they put all together $\sum_{t \in T} \mathbf{P}(a(t)=j) \le \frac{24}{\Delta_i^{\prime 2}}+\underbrace{\sum_{l \geq 8 / \Delta_i^{\prime}}} \Theta\left(e^{-\Delta_i^{\prime 2} l / 2}+\frac{1}{(l+1) \Delta_i^{\prime 2}} e^{-D_i l}+\frac{1}{e^{\Delta_i^{\prime 2} l / 4}-1}\right)+L_i(T)+1+\frac{1}{d\left(x_i, \mu_i\right)}+1 $, the highlighted index of the summation runs on every possible number of plays of the optimal arm and not of arm $j$, so I think the bound you have proposed would hold either for $\sum_{t \in T} \mathbf{P}(a(t)=j, n_j(t)> L_{i,j}(T), n_1(t)>L_{i,j}(T))$ or $\sum_{t \in T} \mathbf{P}(a(t)=j, n_j(t)> L_{i,j}(T), E)$, where is an additional event occurring for the sample of the $j-th$ arm, am I missing something? I thank the authors for their willingness to address potential issues in the writing and for the enthusiasm with which they engaged in the discussion with the reviewers.

---

> > > ### Author Response · Authors · 2025-08-27
> > > **Reply to Reviewer ywM1**
> > >
> > > Thank you for the follow-up! We’re not entirely sure we fully understood your point — the markdown rendering may have caused some confusion, so please let us know if we missed anything.
> > >
> > > As you mentioned, the summation term is indeed indexed by the sample count of the optimal arm instead of the sub-optimal arm. That is correct. However, the key fact is that this term can be upper bounded by a constant: it is **a sum of exponentially decaying terms** in the number of optimal arm plays, which is known to be of order $O(1).$ This means, there exists a constant  $C$, that can replace the summation term. This conclusion can be also reflected in the proof of Theorem 1.1 of Agrawal (see page A 12 in https://www.columbia.edu/~sa3305/papers/j3-corrected.pdf). This is why our result does not include an additional term growing with the number of optimal arm plays or have an additional conditional term.

---

> > > > ### Comment · Reviewer_ywM1 · 2025-08-27
> > > >
> > > > Thank you again for the kind and clear reply, if I have understood correctly you make the same steps considering the contributions $\propto O(\frac{1}{\Delta'^{4}})$ as $O(1)$ as the these terms in such a setting are constant w.r.t. to the time horizon $T$. If this is the case I think you are right and the argument holds, however I would suggest to include a little proof to help navigate the sense of the lemma. I again thank the authors for their answers and I hope my observations were helpful to find the writing issues and correct the proofs, I know think there is now enough mathematical evidence to support the claims. I would like to raise one last point (that doesn't impact my judgement on the paper as it is a sophistication). From my understanding both $\mathcal{T}_{k}^{a,o}(t)$  and are sets made of random variables as those set are dependent on the realization, so that (little before equation 4) in $E[N{i,j} (T)]=E[\sum _ { t \in \mathcal{T} _{k} ^{a,o}(t)} 1(I_t=j)]$ you cannot trivially put the expectation value acting on the summands obtaining  $E[N{i,j} (T)]=\sum _ { t \in \mathcal{T} _{k} ^{a,o}(t)} Pr(I_t=j)$, but if I get it correctly you should consider $E[N{i,j} (T)]=E[\sum _ { t \in \mathcal{T} _{k} ^{a,o}(t)} Pr(I_t=j)|  \mathcal{T} _{k} ^{a,o}(t)]$, however is not much of a concern for me because I think all the results recovered by the authors hold for every realization of the sets independently from the specific realization of the sets itself. I thank again the authors for their willingess to engage in the discussion and I'm sorry if I have changed a little bit the notation in order to make it possible to generate the text without errors.

---

> > > > > ### Author Response · Authors · 2025-08-28
> > > > > **Reply to ywM1**
> > > > >
> > > > > We really appreciate your careful reading — your comments are very helpful for improving both the writing and the correctness of our analysis. We will include these discussions in the final submission and, as you suggested, provide more details in the proof of the lemma.
> > > > >
> > > > > Regarding your last point: yes, as you noted, we should use the expectation form to make the expression fully rigorous. Since the probability term is upper bounded for every possible $\mathcal{T} _k^{a,o}(t),$ we can still derive the same result. We will incorporate this correction in the final version. Thank you again for your efforts in helping us strengthen the proof!

---

> > > > > > ### Comment · Reviewer_ywM1 · 2025-08-28
> > > > > >
> > > > > > Thank you again, I consider myself satisfied with your answers.

---

### Review · Reviewer_tro3 · 2025-08-19

**Summary Of Contributions:**

The paper tackles the case of bandits with an unknown cool-down period after selecting a given arm. More specifically, the rewards follow a Bernoulli distribution, with an additional cool-down. After introducing the problem and giving intuition on the main difficulties around it, it presents novel algorithms for both known and unknown cool-down cases, before deriving regret bounds for each and comparing them to the sleeping bandits case. The paper ends with some numerical experiments and additional insights.

**Audience:**

Yes

**Broader Impact Concerns:**

None.

**Claims And Evidence:**

Yes

**Requested Changes:**

See weaknesses section. Other smaller items:
- More context and the use cases of such framework
- Limitations discussion

**Strengths And Weaknesses:**

Strengths:
- The problem tackled (case of unknown active status of arms) is interesting and promising
- Theoretical analysis is rigorous giving a strong mathematical grounding to the results presented
- Paper is clearly written and reasoning and well explained, with interesting intuitions on the problem in section 3 and insights in section 6

Weaknesses:
I believe the main limitation of the paper is the simplicity of the setting it chose: having a 1-0 reward seems rather limiting and I believe a lot of valuable insights would come from the case of slightly more complex rewards (e.g. trade-off between high reward and cool-down).

---

> ### Author Response · Authors · 2025-08-26
> **Reply to Reviewer tro3**
>
> Thank you for your positive feedback! To answer your question regarding reward distribution, let us recall, the major challenge resulted from the cool-down period is that，when you receive a 0 reward, it is hard to tell whether it is really a reward or a penalty due to selecting an inactive arm. This means, cool-down period raises a challenging issue when and only when the probability of reward being 0 is positive. This is why we choose to discuss Bernoulli distribution in the paper: it is a most general distribution that meets the requirement. Besides, Bernoulli distribution is also a popular setting in bandit learning framework, especially those studying Thompson sampling algorithms. You mentioned "trade-off between high reward and cool-down" might be more insightful. Actually in bandit setting, "reward mean" is what we consider, and our Bernoulli setting can also reflect the trade-off (the reward mean in Bernoulli distribution is equivalent to the probability that reward is 1).
>
> Then go back to your question, is it possible the reward setting is more complicated? The answer is YES! Our algorithms can be roughly divided into two parts: Thompson Sampling framework deals with the estimate of the reward mean and exploiting, and the cool-down exploration deals with the estimate of the cool-down duration. In this sense, we only need to "update" the Thompson sampling rule in our algorithm to be able to deal with more general reward distribution. For example, by changing our Algorithm 3 to Algorithm 2 in https://arxiv.org/pdf/1111.1797, we can deal with general stochastic reward distributions. As for the theoretical results, we only need to change the analysis details regarding Thompson Sampling while persisting the main analysis flow. We will add the above discussion in the final submission.
>
> Other example for our setting includes some financial trading/gambling, where we have trading intervals, and clinical trails, where an interval is also needed. As for other limitation, our problem setting does not extend to stochastic cool-down period. When the cool-down duration is stochastic and undisclosed to the agent, in general we can expect an algorithm-independent linear regret. This is because even if the agent is aware of the distributions of these durations, the likelihood that the estimated cool-down duration differs from the actual value at each time step remains significantly high (constantly large). Consequently, the probability that the agent selects a sub-optimal arm at each time instance does not converge to 0 (remains a positive constant). This leads to a linear regret regardless of the algorithm to be used.

---

### Decision · Action_Editor_2pGB · 2025-10-07

**Recommendation:** Accept with minor revision

**Additional Comments:**

The authors are encouraged to revise the paper in light of the discussion. In particular, they should fix the proofs as agreed during the discussion.

**Audience:**

Yes

**Audience Explanation:**

The paper presents a setting that may be of interest to a portion of the community.

**Claims And Evidence:**

Yes

**Claims Explanation:**

Although the reviewers had concerns about the correctness of some of the proofs, the subsequent discussion succeeded in clarifying and resolving those issues.

---

> ### Author Response · Authors · 2025-11-03
> **Reply to Action Editor 2pGB**
>
> We have uploaded a camera ready revision that fixed the statement of Lemma 3 and the analysis of Theorem 2. In addition, we added a discussion in Section 6 as the reviewer requested.